# Cooperation in a Transboundary River Basin: a Large Scale Socio-hydrological Model of the Eastern Nile

Mohammad Ghoreishi[1,2], Amin Elshorbagy[2,3,4], Saman Razavi[1,2,3,5], Günter Blöschl[6], Murugesu Sivapalan[7], and Ahmed Abdelkader[3]

[1]School of Environment and Sustainability, University of Saskatchewan, Canada

[2]Global Institute for Water Security, University of Saskatchewan, Saskatoon, Saskatchewan, Canada

[3]Department of Civil, Geological, and Environmental Engineering, University of Saskatchewan, Canada.

[4]International Institute for Applied Systems Analysis (IIASA), Laxenburg, Austria

[5]Institue for Water Futures, Mathematical Science Institute, Australian National University, Canberra, Australia

[6]Centre for Water Resource Systems, Vienna University of Technology, Austria

[7]Department of Civil and Environmental Engineering, the University of Illinois at Urbana-Champaign, USA

*Correspondence to*: Mohammad Ghoreishi (Mohammad.ghoreishi@usask.ca)

## Abstract

While conflict-and-cooperation phenomena in transboundary basins have been widely studied, much less work has been devoted to representing the process interactions in a quantitative way. This paper identifies the main factors in the riparian countries' willingness to cooperate in the Eastern Nile River Basin, involving Ethiopia, Sudan, and Egypt, from 1983 to 2016. We propose a quantitative model of the willingness to cooperate at the national and river basin scales. Our results suggest that relative political stability and foreign direct investment can explain Ethiopia's

decreasing willingness to cooperate between 2009 and 2016. Further, we show that the 2008 food crisis may account for Sudan recovering its willingness to cooperate with Ethiopia. Long-term lack of trust among the riparian countries may have reduced basin-wide cooperation. While the proposed model has some limitations regarding model assumptions and parameters, it does provide a quantitative representation of the evolution of cooperation pathways among the riparian countries, which can be used to explore the effects of changes in future dam operation and other

management decisions on the emergence of conflict and cooperation in the basin.

## 1 Introduction

Managing transboundary rivers is a complex issue as riparian countries with different socio-economic statuses and interests share water resources. The contested use of these shared water resources can lead to water conflicts (i.e., a

dispute between countries over the rights to water resources) or cooperative agreements (i.e., the peaceful management and use of water resources by the various riparian countries) (Wolf et al., 2003; Zeitoun & Mirumachi, 2008). An important issue in managing transboundary water resources is to explain why Conflict and Cooperation (C&C) emerge in a transboundary river and investigate its potential feedback mechanism. Conflicts may emerge as self-interested riparian countries maximize their benefits by using shared water resources, while long-term benefits of cooperative

agreements amongst riparian countries can lead to cooperation (Ball, 2012). A good understanding of how C&C come about requires analysis of the relevant socio-hydrological factors in the decision making and their interactions over time (Lu et al., 2021; Sivapalan & Blöschl, 2015; Wei et al., 2019). This understanding can provide informative insights on how to avoid conflicts and improve cooperation in a transboundary river basin.

Zeitoun and Warner (2006) developed a conceptual framework of hydro-hegemony, which results from a riparian

country's upstream position or historical water rights. In this study, hydro-hegemony is defined as the leadership or dominance of a riparian country over other riparian countries in a transboundary river basin due to a riparian country's upstream position or historical water rights. Water management decisions (e.g., operating rules and dam construction) of upstream countries with socio-economic benefits for different sectors, such as agriculture and industry, often negatively influence downstream flows, which can motivate downstream countries to negotiate with upstream

countries to increase their own socio-economic benefits (Nicol & Cascão, 2011). Such negotiations can lead to different levels of cooperation (e.g., a treaty signed by riparian countries) (Zeitoun & Mirumachi, 2008), which in turn can change water use and/or dam operating rules or result in dam construction along the river basin (Cascão, 2009).

The transboundary rivers have been receiving significant attention in many studies (e.g., Elhance, 1999; Kilgour & Dinar, 2001; Wolf, 2007). The literature of transboundary rivers has generally focused on pathways towards resolving

conflicts (e.g., Madani et al., 2014; Rogers, 1969; Zarezadeh et al., 2012), analyzing C&C (e.g., Mirumachi & Van Wyk, 2010; Wolf, 2007; Wolf et al., 2003), and investigating influential factors in C&C (e.g., Dinar et al., 2010; Zeitoun et al., 2011), often in a scenario-based context. Recently, C&C in transboundary water systems has attracted the attention of socio-hydrological research, which focuses on the coevolutionary behavior of social and hydrological systems (Sivapalan et al., 2012). The endogeneity of humans in water systems has been the subject of numerous socio-

hydrological studies using a variety of methods, including those incorporating socio-economic drivers (Aghaie et al., 2020; Elshafei et al., 2014), or those employing concepts of social memory (Di Baldassarre et al., 2013; Gonzales & Ajami, 2017a) and collective behavior (Du et al., 2017; Garcia et al., 2016). Compared to other studies on transboundary rivers, socio-hydrological research emphasizes quantifying C&C dynamics by including both socio-political and hydrological factors in modeling (e.g., Lu et al., 2021) and providing a general framework on C&C (e.g.,

Wei et al., 2022). However, previous studies on socio-hydrology in transboundary rivers did not focus on the important concept of social memory and quantitative components of C&C phenomena (e.g., political stability). Also, the advantage of qualitative data and narratives for model validation is elusive in the current socio-hydrological research on transboundary rivers. Thus, more research needs to be conducted to understand C&C in other transboundary rivers and investigate the associated socio-political factors with the use of qualitative data for model validation. This study

is intended to contribute toward filling some of these research gaps.

As a transboundary river, the Nile River Basin has a long history of cooperation and conflict (Cascão, 2009; Nicol & Cascão, 2011). Water conflicts are more severe in the Eastern Nile Basin (ENB), where Ethiopia, Sudan, and Egypt are located, due to water scarcity. A recent dispute arose when Ethiopia announced the Grand Ethiopian Renaissance Dam (GERD) construction in 2011, which would reduce the water released to Sudan and Egypt during the filling

period (Whittington et al., 2014), and later because of increased evaporation from the reservoir surface. Also, GERD operation is anticipated to smooth the peaks of the Blue Nile flow and increase the low flows in the spring period (Abdul Latif Jameel Water and Food Systems Lab, 2014) which may be positive for Sudan as it lacks any multi-year water storage facilities. On the other hand, Egypt is more concerned about the annual inflow volume into the country as the High Aswan Dam (HAD) can buffer seasonal fluctuations due to its large storage capacity (Abdul Latif Jameel

Water and Food Systems Lab, 2014), so the negative effects of the GERD are perceived to dominate. The conflict over the GERD has been aggravated over the past several years. Importantly, the conflict among the three countries has been attracting significant international attention, and therefore, we focused on the ENB in this study, which can contribute to addressing large issues across the entire Nile Basin.

Previous studies on the Nile River Basin have addressed the trade-offs among riparian interests and analyzed potential

developments in the basin (Arjoon et al., 2014; Block & Strzepek, 2010; Digna et al., 2018; Geressu & Harou, 2015; Jeuland & Whittington, 2014; Kahsay et al., 2015; Nigatu & Dinar, 2016; Sangiorgio & Guariso, 2018; Strzepek et al., 2008; Wheeler et al., 2018, 2020, 2016). These studies are mostly based on multi-objective optimization to quantify optimum trade-offs among conflicting interests (Geressu & Harou, 2015; Wheeler et al., 2018), or scenario analyses to explore potential alternatives for basin management (Mulat & Moges, 2014a; Wheeler et al., 2016). Socio-

hydrological modeling on the other hand can provide an explanation of how cooperation emerges in the Eastern Nile River Basin as the result of the interplay of hydrological and socio-political factors (Ghoreishi, Sheikholeslami, et al., 2021; Lu et al., 2021).

This study builds on the previous work on socio-hydrology in transboundary rivers by Lu et al. (2021) and Wei et al.
(2022) but distinguishes itself by incorporating the following additional important elements into our socio-hydrologic
model: (1) the concept of social memory and quantitative components of C&C phenomena (e.g., political stability),
(2) uncertainties in the representation of countries' decision making process, and (3) the heterogeneity of decision
making across the riparian countries in their cooperation.The aim of this paper is therefore to quantify feedback
mechanisms in the cooperation within the ENB by: (1) investigating the socio-economic and political factors in the
95 riparian countries' willingness to cooperate and (2) quantifying and simulating the riparian countries' willingness to
cooperate resulting from the interactions between these various factors. By developing a socio-hydrological model,
we provide a quantitative explanation of how hydrological and socio-political factors may have led to cooperation and
conflict in the ENB from 1983 to 2016. The simulations are validated with qualitative data and narratives in the basin
to demonstrate their plausibility.

**2 Historical Events of Conflict and Cooperation in the Nile River Basin – Case study**

The Nile River is the longest transboundary river in the world, with the main stem of $6,695 \times 10^3$ m. This river basin is
shared among 11 countries: Tanzania, Uganda, Rwanda, Burundi, The Democratic Republic of the Congo, Kenya,
Ethiopia, Eritrea, South Sudan, Sudan, and Egypt. The river has two main tributaries, the White Nile and the Blue
Nile (Abay) (Figure 1), each with distinct hydrologic regimes. The ENB (Figure 1), covering the Blue Nile, Atbara
(Tekezze), and the Baro Rivers, contributes 85–90% of the Nile River water flowing into Egypt with high seasonality;
the White Nile contributes the remains of the water with a more uniform streamflow.

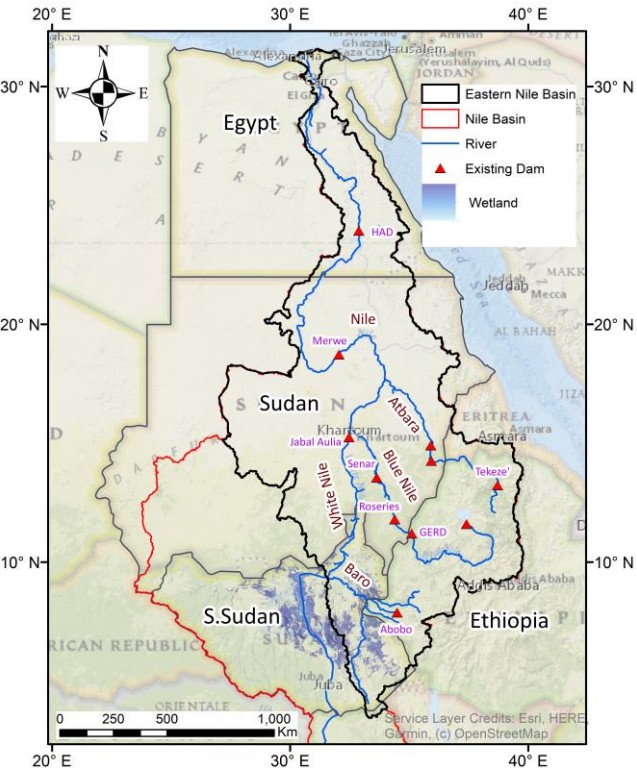

**Figure 1: The Nile River Basin (red line), and the Eastern Nile Basin (black line) with the three main riparian countries
(Egypt, Sudan, and Ethiopia) (retrieved from Esri (2021))**

Figure 2 indicates the main cooperation and conflict events from 1959 to 2020 among the riparian countries in the Nile River Basin. This time period is longer than that of our modeling in section 3 as we need a long time period to investigate the socio-economic and political factors in the riparian countries' willingness to cooperate. It is worth
mentioning that we did not use this long period for section 3 due to the lack of quantitative data. Under the 1959 treaty, Egypt and Sudan agreed on the use of the Nile River water — an agreement that excluded Ethiopia (Food and Agriculture Organization, 1959) and other riparian countries. This agreement was followed by the construction of the Roseires Dam and the HAD in 1966 and 1970, respectively, both multipurpose structures (Food and Agriculture Organization, 1959) (Figure 1). The 1959 treaty shows the important role of dam constructions as a motivation for
countries to cooperate. The 1968–1973 drought and food security issue in Ethiopia heightened awareness among Ethiopians of the importance of the Nile water and the need to negotiate with the other riparian countries (Nicol & Cascão, 2011). In 1983, several countries in the basin formed the Undugu group to achieve regional cooperation, but Kenya, Tanzania and Ethiopia joined only as observers (Kagwanja, 2007; Seide, 2014). In 1992, the riparian countries formed the Technical Cooperation Commission for the Promotion and Development of the Nile (TECCONILE) to
pave the way for the guidelines for the Nile Basin Initiative (NBI) (Kagwanja, 2007; Seide, 2014). Signed in 1999 by nine countries—Egypt, Sudan, Ethiopia, Uganda, Kenya, Tanzania, Burundi, Rwanda, and the Democratic Republic of Congo—the NBI was intended "to achieve sustainable socio-economic development through the equitable utilization of and benefits from the shared Nile Basin water resources" (NBI, 2019). This partnership was followed by an agreement termed the Joint Multipurpose Project (JMP) in 2003 for widespread economic improvements along
the river basin.

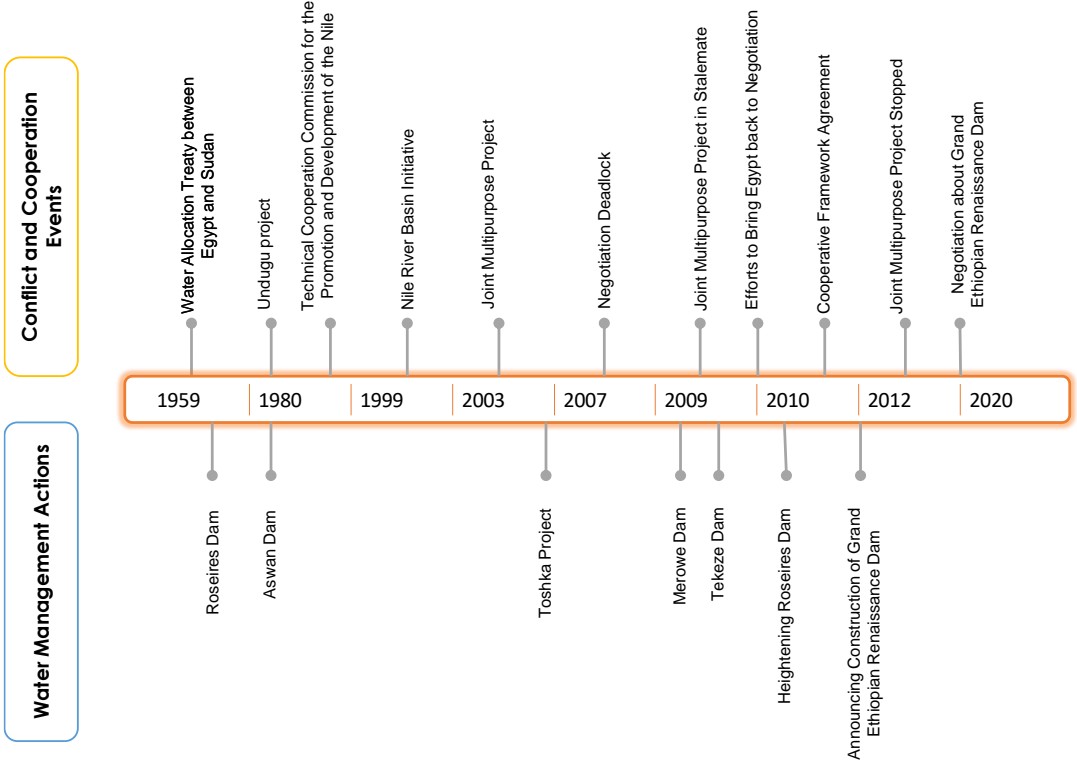

**Figure 2: Timeline of conflict and cooperation events and water management actions in the Nile River Basin, with a particular focus on the three riparian countries of Egypt, Sudan, and Ethiopia**

However, these cooperative agreements did not prevent countries from making unilateral developments. The irrigated
areas extended rapidly in Egypt with significant foreign direct investments. In 2005, Egypt launched an important part of the Toshka project (Nicol & Cascão, 2011), transferring water to irrigate a part of the Western Desert of Egypt to expand the agricultural area. Ethiopia built the Tekeze Dam, with a capacity of $9.3 \times 10^9$ m$^3$, on the Atbara River in

2009 (Figure 1) for hydropower generation (Cascão, 2009). Sudan completed the Merowe Dam, with a capacity of $12.5 \times 10^9$ m$^3$, on the main Nile River (Figure 1), again for hydropower purposes (Cascão, 2009).

The countries' negotiations passed through many different stages, and a Cooperative Framework Agreement (CFA) was about to be reached in 2007, but representatives from Sudan and Egypt expressed strong disagreement about a part of this agreement because they were concerned about possible negative impacts on their historical water rights (Nicol & Cascão, 2011). Their objections led to a negotiation deadlock in 2007 (See Article 14b in the Annex of CFA (2010)) which weakened the high expectations for the JMP and resulted in a stalemate in 2009. Attempts to bring 145 Egypt back to the negotiating table were unsuccessful, and the upstream countries without Burundi, the Democratic Republic of Congo, Sudan, and Egypt reached a CFA in 2010. However, as a two-thirds majority of all riparian countries was required, the agreement was not ratified. As a reaction to CFA, Sudan and Egypt froze all their participation in the NBI (Nicol & Cascão, 2011). Also, Egypt convinced Burundi and the Democratic Republic of Congo to not sign the agreement (Nicol & Cascão, 2011).

Between 2010 and 2012, the Arab Spring created political upheaval in Egypt (Knaepen & Byiers, 2017; Salman, 2013). Meanwhile, the global food crisis of 2008 and South Sudan's independence from Sudan in 2011 convinced Sudan to refocus on the agricultural sector, for which it received foreign direct investments between 2008 and 2012. To proceed with an ambitious agricultural improvement plan, Sudan started raising the height of the Roseires Dam in 2010, and the country rejoined the NBI in 2012 (Cascão & Nicol, 2016; Nicol & Cascão, 2011). Ethiopia, given its 155 improved political stability, the changed situation in the basin, and the substantial foreign direct investments, announced the construction of GERD, with a total storage capacity of $74 \times 10^9$ m$^3$ (Cascão & Nicol, 2016). The dam was perceived as a water threat by Egypt and possibly Sudan (Abdul Latif Jameel Water and Food Systems Lab, 2014) so, although the JMP did not continue because of reduced financial support, negotiations among Egypt, Sudan, and Ethiopia were ongoing (Cascão & Nicol, 2016), resulting in signing a declaration on GERD in 2015 (Agreement on 160 Declaration of Principles between Egypt, Ethiopia, and Sudan, 2015). According to this declaration, Ethiopia, Sudan, and Egypt would agree on a guideline for the GERD filling and operation phases, with the dam owner reserving the right to adjust the rules from time to time, informing downstream countries of the situation leading to this adjustment. Since then, growing concerns about the negative impacts of the GERD on the downstream countries have increased the disagreements among Egypt, Sudan, and Ethiopia (Abdul Latif Jameel Water and Food Systems Lab, 2014; Kahsay 165 et al., 2017; Mulat & Moges, 2014b; Wheeler, 2017).

**3 Methods**

In section 2, we initially detected the most important socio-political and hydrological factors in C&C dynamics in the ENB through the existing narrative. These factors are taken into account in section 3.1 where we fully define all these 170 factors and validate their significant role in ENB's C&C dynamics with local and general studies in transboundary rivers. In section 3.2, we conceptualize the interactions of the C&C dynamics in the basin based on the ENB literature. Spatially, each of Ethiopia, Sudan, and Egypt is considered as one unit, and temporally we adopt an annual resolution. In section 3.3, we introduce the hypothesized equations to simulate the riparian countries' willingness to cooperate and the basin-wide cooperation. All model inputs and the setting of sensitivity analysis are introduced in sections 3.4 175 and 3.5, respectively.

**3.1 Variables**

In the first step, we define input variables and model variables (Table 1). The input variables are reservoir storage under construction, food gap, energy gap, relative political stability, and foreign direct investment.

**Future reservoir storage capacity (S/PDC):** The Future reservoir storage capacity is defined as the ratio of Reservoir 180 storage under construction to potential dam capacity. According to the existing Nile plans, the hydropower capacities are planned to increase to 26,000 MW in the basin by 2050 (NBI, 2016). The prospects of increasing reservoir storage both for irrigation and hydropower may enhance C&C in transboundary rivers (Food and Agriculture Organization, 1959).

**Food gap (FC/FP):** The food gap is defined as the ratio of food consumption to the produced food from rainfed and irrigated areas. The food gap has been recognized as an important factor in the C&C within the Nile River Basin (Kagwanja, 2007; Nicol & Cascão, 2011), which can motivate riparian countries to cooperate on agricultural developments. Food production in each country is affected by water allocation within a country and transboundary flow from other countries.

**Energy gap (PEC /HP):** The energy gap is defined as the ratio of potential energy capacity to actual hydropower generation. Because the Nile River Basin has a high potential power capacity, several hydropower plants have been proposed under unilateral/multilateral agreements, in addition to existing hydropower plants (NBI, 2016). The energy gap can also be increased or decreased by changing reservoir operation rules (Basheer et al., 2018; Wheeler et al., 2018) and can impact C&C through energy export. Energy production in each country is affected by water allocation within a country and transboundary flow from other countries.

**Relative political stability (PS):** Political stability has been recognized as an important factor in countries' willingness to cooperate in decisions about transboundary rivers (Zeitoun & Warner, 2006; Zeitoun et al., 2017) and the Nile River Basin (Cascão, 2009; Di Nunzio, 2013). A country's political stability index represents indicators of political stability, such as government stability and the presence of protest or civil war (Worldwide Governance Indicators, 2021). In this study, the relative political stability is estimated by normalizing the data to the range between 0.1 and 0.9, where 0.1 and 0.9 represent the least and the most stable states, respectively (see section 3.4 for more details).

**Foreign direct investment (FDI):** Foreign direct investment is an investment in a country by another country (World Bank, 2021). Several qualitative studies have found that FDI influences C&C in transboundary rivers, including the Nile Basin (Cascão, 2009; Waterbury, 2008; Whittington et al., 2014; Zeitoun & Mirumachi, 2008; Zeitoun & Warner, 2006; Zeitoun et al., 2011, 2017). In this study, FDI is normalized relative to the maximum historical value to range between 0.1 and 0.9, where 0.1 and 0.9 represent the smallest and the largest FDI (see section 3.4 for more details).

The model variables are memory of cooperation, willingness to cooperate, and cooperation in the ENB.

**Memory of cooperation (CS):** A countries' memory of cooperation is an indicator of the historical basin cooperation on the willingness to cooperate. Past studies have suggested the important role of memory and trust in C&C in transboundary rivers, including the Nile River Basin (Cascão, 2009; Metawie, 2004; Tafesse, 2001; Whittington et al., 2014; Zeitoun & Mirumachi, 2008). Here, memory is defined as a weighted average of the countries' historical willingness to cooperate over their memory span.

**Willingness to cooperate (WC):** A countries' willingness to cooperate is an indicator of how likely the country is to change its dam operations or build a reservoir for multilateral projects. The countries' willingness to cooperate is the main driver of the basin-wide C&C.

**Cooperation in the ENB (BC):** The cooperation in the ENB is a variable that reflects the overall level of cooperation and is estimated as the average of willingness to cooperate of the three countries.

**Table 1: Input variables and model variables. All variables have an annual resolution (with the exception of potential energy capacity and potential dam capacity which are single values) and are specified for Ethiopia, Sudan, and Egypt independently.**

| Input or Model Variable | Symbol | Variables | Units | Estimation, source |
|---|---|---|---|---|
| Input | S/PDC | Future reservoir storage capacity = (Reservoir storage under construction / potential dam capacity) | $m^3 / m^3$ | (NBI, 2016) |

| | | | | |
|---|---|---|---|---|
| Input | FC/FP | Food gap = (Food consumption / food production) | kg/kg | Water resources model: (Abdelkader & Elshorbagy, 2021) |
| Input | PEC/HP | Energy gap = (potential energy capacity / Energy production) | MW/MW | Water resources model: (Abdelkader & Elshorbagy, 2021) / (NBI, 2012) |
| Input | PS | Relative political stability | Unitless | (Food and Agriculture Organization Data, 2020b) |
| Input | FDI | Foreign direct investment | USD | (Food and Agriculture Organization Data, 2020b) |
| Model | CS | Memory of cooperation | Unitless | Eq. 7 |
| Model | WC | Willingness to cooperate | Unitless | Eq. 1, 2, 3 |
| Model | BC | Cooperation in the ENB | Unitless | Eq. 8 |

### 3.2 Conceptualization of system interactions

Figure 3 shows our Eastern Nile Basin Socio-hydrological (ENSH) model conceptualization as a causal feedback diagram.

The willingness to cooperate within each country is controlled by a number of factors. Future reservoir storage capacity, i.e. the ratio of Reservoir storage under construction and potential dam capacity, is assumed to increase the willingness to cooperate, both in Ethiopia and in Sudan ("+"sign in Figure 3). The Roseires and the Aswan High Dams were built as a result of the 1959 treaty in Sudan and Egypt, respectively (Food and Agriculture Organization, 1959). Thus, the desire to increase reservoir storage in Sudan and Ethiopia through multilateral projects, like the Roseires Dam and the HAD, can increase the countries' willingness to cooperate in the basin (Food and Agriculture Organization, 1959).

An increase in Ethiopia's food gap may reduce Ethiopia's willingness to cooperate ("–"sign in Figure 3) (Basheer et al., 2018; Kagwanja, 2007; Nicol & Cascão, 2011; Wheeler et al., 2018). For example, Ethiopia's population growth and food security issues may motivate Ethiopia to use water as an upstream country in the ENB to increase food production (Nicol & Cascão, 2011). The food gap in Sudan is assumed to increase willingness to cooperate ("+"sign in Figure 3). Although Sudan stopped participating in the NBI in 2007, the 2008 food crisis in Sudan and its agricultural development plans later convinced this country to return to the NBI (Cascão & Nicol, 2016; Nicol & Cascão, 2011). Finally, the food gap in Egypt is assumed to increase willingness to cooperate ("+"sign in Figure 3). In other words, Egypt's food gap can convince this country to negotiate with other riparian countries to achieve water development plans that increase the country's food production (Nicol & Cascão, 2011).

An increase in Ethiopia's energy gap is assumed to reduce willingness to cooperate ("–"sign in Figure 3) as reflected by the construction of the GERD, fueling concerns in the basin (Cascão & Nicol, 2016). However, in Sudan, an increase in energy gap or dam constructions can also be a motivation for cooperation with other countries ("+"sign in Figure 3) (Basheer et al., 2018; Wheeler et al., 2018) because basin cooperation may convince Ethiopia to release more water, so Sudan can decrease its energy gap.. Hydropower production in Egypt is not considered here, as it only contributes in a minor way to Egypt's total energy production (NBI, 2016).

An increase in relative political stability is conceptualized to reduce a country's willingness to cooperate ("–"sign in Figure 3). It was reported that the low political stability of Sudan after South Sudan's independence in 2011 encouraged ambitious water development plans (Nicol & Cascão, 2011). After 2010, Egypt's political stability dramatically declined which, together with the low stability and the increasing stability in Ethiopia after 2010, paved the way for Ethiopia's GERD announcement (Nasr & Neef, 2016).

An increase in foreign direct investments is likewise conceptualized to reduce a country's willingness to cooperate ("–"sign in Figure 3). It was proposed that low foreign direct investments motivated Ethiopia to embrace the NBI in the 1990s (Zeitoun et al., 2011). With increasing foreign direct investments after 2011, Ethiopia's willingness to cooperate with Sudan and Egypt declined (Cascão & Nicol, 2016). Similarly, it was reported that, in 2007, a year in

which Egypt enjoyed a high level of foreign direct investments, the country had disagreements with other riparian countries, but in 2015, when Egypt's foreign direct investments were relatively low, it was more likely to cooperate (Cascão & Nicol, 2016).

Memory of cooperation (CS) is assumed to increase the willingness to cooperate. Positive memories of basin cooperation can lead to trust and an improved environment in the basin (Whittington et al., 2014). While the 1959 treaty between Egypt and Sudan, for example, triggered feelings of profound mistrust among other riparian countries in the Basin (Whittington et al., 2014), the NBI provided a unique opportunity for the riparian countries to improve their cooperative plans and build trust in the basin (Metawie, 2004).

The three countries, Ethiopia, Sudan, and Egypt are connected in two ways: First, the food and energy productions, and thus the food and energy gaps, in each country are affected by water allocation within a country and transboundary flow from other countries with are estimated by a water management model (see section 3.4). The transboundary interactions are thus embedded in the food and energy gaps and not depicted explicitly in Figure 3. Transboundary flow is the river flow minus the water withdrawals of different water uses, using the water management model (further

details in section 3.4). Decreasing transboundary flow from Ethiopia reduces energy/food production downstream, thereby, increasing Sudan's and Egypt's willingness to cooperate within the ENB

Second, the cooperation in the ENB is linked to the willingness to cooperate in each country in a bidirectional way. Each country's willingness to cooperate increases the overall cooperation in the ENB ("+"sign in Figure 3). In the ENB, it was reported that each country's willingness to cooperate increased the probability of basin cooperation in

the past (Cascão & Nicol, 2016). On the other hand, each country's willingness to cooperate is increased by the overall cooperation in the ENB ("+"sign in Figure 3) through a positive environment and trust in the basin. It was proposed that positive memories of basin cooperation can lead to trust and an improved environment in the basin, which can increase countries' willingness to cooperate (Whittington et al., 2014). Thus, for each country, a feedback loop can be seen between willingness to cooperate, memory of cooperation, and cooperation in the basin. This reinforcing loop

implies that a change (i.e., increase or decrease) in one variable is compounded by more change.

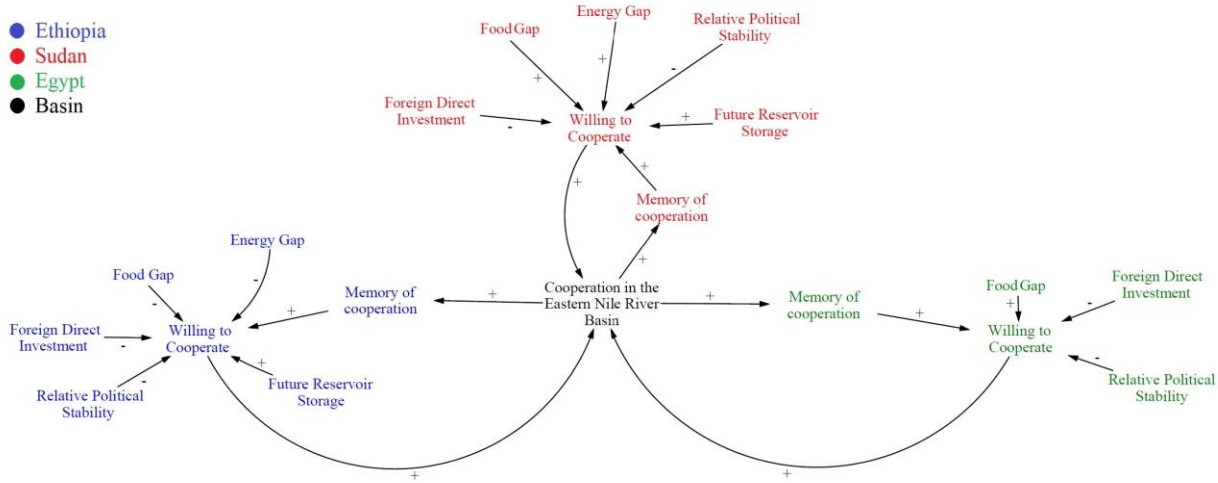

**Figure 3: Conceptualization of the Eastern Nile Basin socio-hydrological (ENSH) model. Input variables and model variables are shown in blue and black, respectively. Signs indicate positive (+) and negative (–) effects. The colours of**

**blue, red, green, and black indicate the variables of Ethiopia, Sudan, Egypt, and basin respectively).**

**3.3 Model Equations**

The causal feedback diagram of Figure 3 is now translated into model equations. The "willingness to cooperate" (WC) of a riparian country is a variable in the range zero (i.e., no cooperation) to one (i.e., perfect cooperation), and we propose Equations 1 to 3 for its estimation. We set up the equations of willingness to cooperate based on human behavior in resource dilemmas. Consistent findings show that decision behaviors in resource dilemmas are mostly individualism (i.e., the drive to prioritize one's own interests) and competition (the drive to increase relative gains, the gap between one's achievement and that of the other) (Brewer & Kramer, 1986; Parks & Vu, 1994; Roch & Samuelson, 1997). Thus, here, countries' willingness to cooperate is evaluated based on their relative socio-economic gains (Kopelman et al., 2002; Mason, 2004).

We assumed that the energy gap, the food gap, and future reservoir storage have additive effects on willingness to cooperate, while the other factors (i.e., relative political stability, foreign direct investment, and countries' memory from past cooperation at the basin) have multiplicative effects. It was reported that our selected multiplicative factors have a strong effect on willingness to cooperate when these factors become extremely low or high compared to the factors with additive effects (e.g., Cascão & Nicol, 2016; Nicol & Cascão, 2011). In other words, the assumption on additive and multiplicative factors means that the effects of a country's energy gap, food gap and future reservoir storage act in a concerted way, while the other variables can each have a stronger impact on how willing riparian countries are to cooperate. Thus, near zero values of multiplicative factors (e.g., a very bitter memory of past cooperation) can make willingness to cooperate near zero even if the other multiplicative variables are much larger.

Ethiopia's willingness to cooperate (Equation 1) is the sum of the inverse food gap, the inverse energy gap and future reservoir storage multiplied by "1 minus relative political stability (*1-PS*)", "1 minus foreign direct investment (*1-FDI*)", and the country's memory. We use the inverse of the food and energy gaps to reflect the negative relationship in the causal loop diagram. The inverse of the gaps was used rather than a negative coefficient because of a more plausible limiting behavior. As relative political stability and foreign direct investment vary from 0 to 1 (i.e., low to high values), subtracting them from 1 produces an inverse effect between these variables and a willingness to cooperate. The country's memory of cooperation positively affects the willingness to cooperate, and it is scaled by the total number of riparian countries, n=3. In Sudan and Egypt (Equations 2 and 3), the food gap has a direct effect on the willingness of both to cooperate, and so has the energy gap in Sudan.

$$WC_{Eth,t} = \left( \beta_{1,Eth} \cdot \left( \frac{FC_{Eth,t}}{FP_{Eth,t}} \right)^{-1} + \beta_{2,Eth} \cdot \left( \frac{PEC_{Eth,t}}{HP_{Eth}} \right)^{-1} + \beta_{3,Eth} \cdot \frac{S_{Eth,t}}{PDC_{Eth}} \right)^{\alpha_{1,Eth}} \cdot \left( 1 - PS_{Eth,t} \right)^{\alpha_{2,Eth}} \cdot \left( 1 - FDI_{Eth,t} \right)^{\alpha_{3,Eth}} \cdot \left( \frac{CS_{Eth,t}}{n} \right)^{\alpha_{4,Eth}} \tag{1}$$

$$WC_{Sud,t} = \left( \beta_{1,Sud} \cdot \frac{FC_{Sud,t}}{FP_{Sud,t}} + \beta_{2,Sud} \cdot \frac{PEC_{Sud}}{HP_{Sud,t}} + \beta_{3,Sud} \cdot \frac{S_{Sud,t}}{PDC_{Sud}} \right)^{\alpha_{1,Sud}} \cdot \left( 1 - PS_{Sud,t} \right)^{\alpha_{2,Sud}} \cdot \left( 1 - FDI_{Sud,t} \right)^{\alpha_{3,Sud}} \cdot \left( \frac{CS_{Sud,t}}{n} \right)^{\alpha_{4,Sud}} \tag{2}$$

$$WC_{Egy,t} = \left( \beta_{1,Egy} \cdot \frac{FC_{Egy,t}}{FP_{Egy,t}} \right)^{\alpha_{1,Egy}} \cdot \left( 1 - PS_{Egy,t} \right)^{\alpha_{2,Egy}} \cdot \left( 1 - FDI_{Egy,t} \right)^{\alpha_{3,Egy}} \cdot \left( \frac{CS_{Egy,t}}{n} \right)^{\alpha_{4,Egy}} \tag{3}$$

with $\alpha > 0$, $\beta > 0$. The subscript *t* represents time. The coefficients $\beta_1$, $\beta_2$, and $\beta_3$ add up to one. They represent the countries' emphasis on the food gap, the energy gap, and future reservoir storage, respectively, in line with the policymakers' preferences. The coefficients for Ethiopia and Sudan are assumed to be 0.33, following the assumption that the countries pay the same attention to the food gap, the energy gap, and future reservoir storage (Mason, 2004). The coefficient for Egypt is assumed to be 1, as it is the only coefficient. The exponents $\alpha_1$, $\alpha_2$, $\alpha_3$, and $\alpha_4$ represent the countries' emphasis on multiplicative variables of decision making, and they all sum up to one. We assumed all multiplicative weights to be 0.25 because these countries are believed to put the same emphasis on all these multiplicative variables based on our understating of the system.

The memory of cooperation *CS* represents a country's prediction for the following year if the other two countries are in a status of cooperation (1) or non-cooperation (0), or in between. *CS* may therefore vary between 0 and 2. Following the El Farol model (Wilensky & Rand, 2015), we used a "bag of strategies" or various possible predictions as the weighted average of the own countries' willingness to cooperate over a fixed number of years or memory span. While past models of human-water systems have often been deterministic (e.g., Di Baldassarre et al., 2013; Viglione et al., 2014) with exceptions (e.g., (Ghoreishi, Razavi, et al., 2021; Viglione et al., 2014)), here we adopt a stochastic approach and allow the weights γ to vary randomly to account for the stochasticity in a country's decision making. "A bag of strategies" for a country can be written as follows (Equation 7):

$$
ST_{k,t} = \begin{cases} ST_{k,1,t} = \gamma_{k,1,1} \cdot \overline{WC}_{-k,t-1} + \gamma_{k,1,2} \cdot \overline{WC}_{-k,t-2} + \cdots + \gamma_{k,1,m} \cdot \overline{WC}_{-k,t-m} \\ ST_{k,2,t} = \gamma_{k,2,1} \cdot \overline{WC}_{-k,t-1} + \gamma_{k,2,2} \cdot \overline{WC}_{-k,t-2} + \cdots + \gamma_{k,2,m} \cdot \overline{WC}_{-k,t-m} \\ \qquad\qquad\qquad\qquad\qquad \vdots \\ ST_{k,i,t} = \gamma_{k,i,1} \cdot \overline{WC}_{-k,t-1} + \gamma_{k,i,2} \cdot \overline{WC}_{-k,t-2} + \cdots + \gamma_{k,i,m} \cdot \overline{WC}_{-k,t-m} \end{cases} \tag{4}
$$

where $S_{k,1,t}$ is the ith strategy of country $k$ included in the bag at time $t$, $\gamma_{k,i,m}$ is the weight for the $i$th strategy at time $t - m$, $\overline{WC}_{-k,t-m}$ is the average of the countries' willingness to cooperate but agent $k$ at time $t - m$, and $t$ is the current year. The memory span ($m$) and the number of strategies ($i$) are both assumed to be 10. The assumption of m=10 means that the riparian countries remember the past events in the basin rather well and is in line with the ENB's narratives and common assumptions in the modeling for cooperation and conflict models (Cascão & Nicol, 2016; Wilensky & Rand, 2015). Each country then selects from the bag the strategy that had provided the best prediction in the previous years (Equation 10), which can be written as follows:

$$
\varepsilon_{k,i,t} = \left| ST_{k,i,t-1} - \overline{WC}_{-k,t-1} \right| + \left| ST_{k,i,t-2} - \overline{WC}_{-k,t-2} \right| + \cdots + \left| ST_{k,i,t-m} - \overline{WC}_{-k,t-m} \right|
$$

$$
\hat{\phi}_{k,t} = \min\{\varepsilon_{k,1,t}, \varepsilon_{k,2,t}, \dots, \varepsilon_{k,i,t}\}
$$

$$
CS_{k,t} = CS(ST_{k,i,t} | \hat{\phi}_{k,t}) \tag{5), (6), (7}
$$

where $\varepsilon_{k,i,t}$ is the residual of the $i$th strategy for country $k$ at time t, and $\hat{\phi}_{k,t}$ is the minimum residual of different strategies included in a bag of strategies for country $k$ at time t. As initial values, we assumed that Ethiopia had a bitter memory of past cooperations (i.e., $CS_{-1:t-m} = 0$) while Egypt and Sudan had positive memories (i.e., $CS_{-1:t-m} = 2$), based on their bilateral projects in the basin (Cascão & Nicol, 2016).

After calculating the willingness to cooperate (WC) of each country, the basin-wide cooperation (BC) is estimated as an average of the countries' WC (Equation 8).

$$
BC_t = \frac{1}{3} \times (WC_{Eth,t} + WC_{Sud,t} + WC_{Egy,t}) \tag{8}
$$

The simulation time window starts in 1983 when the Undugu project was completed (Kagwanja, 2007; Seide, 2014). Based on this cooperative project, we assumed 0.6 for all countries' willingness to cooperate as an initial value.

### 3.4 Model Inputs

Part of the model inputs (future reservoir storage capacity, relative political stability, foreign direct investment, Table 1) is time series taken from the literature.

Future water storage capacity is estimated as the ratio of the reservoir storage under construction and potential dam capacity. The Abodo Dam and Tekeze dams were constructed in Ethiopia in 1985 and 2009, respectively. In Sudan, the Merowe Dam was constructed in 2009, and the Roseires Dam was heightened to increase its capacity from 3 to $7.3 \times 10^9$ m³ in 2013. The potential dam capacity in Ethiopia is $222 \times 10^9$ m³ in the ENB while dam capacity in Sudan and Egypt would be almost zero with the full operation of the GERD (Berhanu et al., 2014).

Political stability, taken from (Food and Agriculture Organization Data, 2020b), is based on the indices given in Table A.1. Figure 4 shows the effect of the Arab Spring on Egypt's reducing political stability in 2011. The figure also indicates the low values of Sudan's political stability related to South Sudan's independence from Sudan in 2011. The relative political stability (PS) is estimated here by normalizing the data in Figure 4 to the range between 0.1 and 0.9, where 0.1 and 0.9 represent the least and the most stable states, respectively. This Political stability data were extended for the years before 2000 using linear extrapolation.

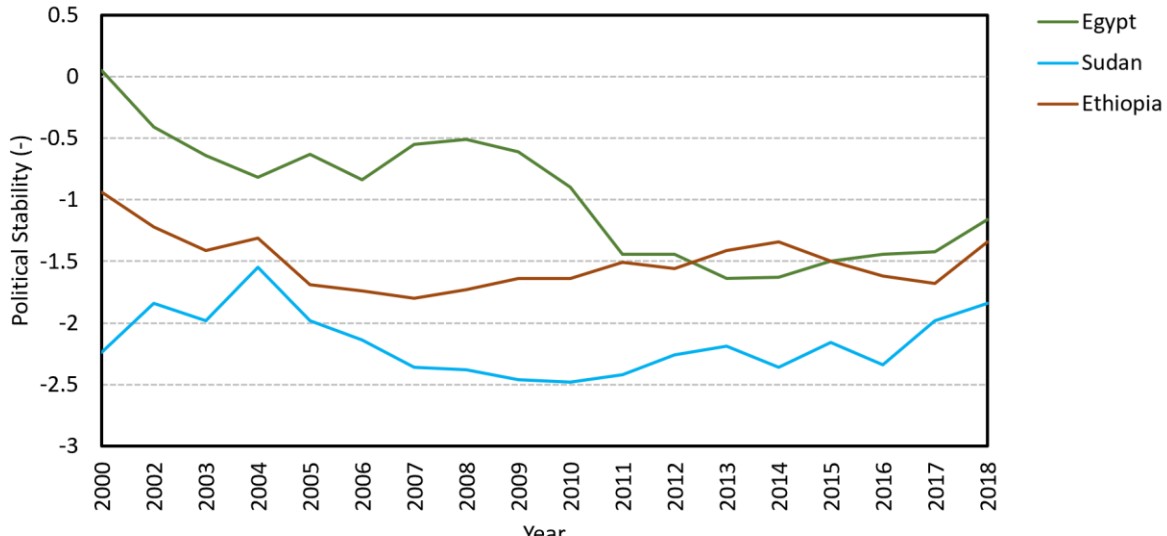

Figure 4: The political stability from 2000 to 2018 of Egypt, Sudan, and Ethiopia (-2.5 weak; 2.5 strong) (Food and Agriculture Organization Data, 2020b)

Foreign direct investment (FDI) is taken from Food and Agriculture Organization Data, (2020b) and shown in Figure 5. After 2003, FDI in Sudan remained almost stable with minor changes. In contrast, in Egypt FDI peaks around 2007, and in Ethiopia there is a clear increase starting in 2012. Foreign direct investment (FDI) normalized here relative to the maximum historical value to range between 0.1 and 0.9, where 0.1 and 0.9 represent the smallest and the largest FDI. The FDI data were extended for the years before 1991 using linear extrapolation.

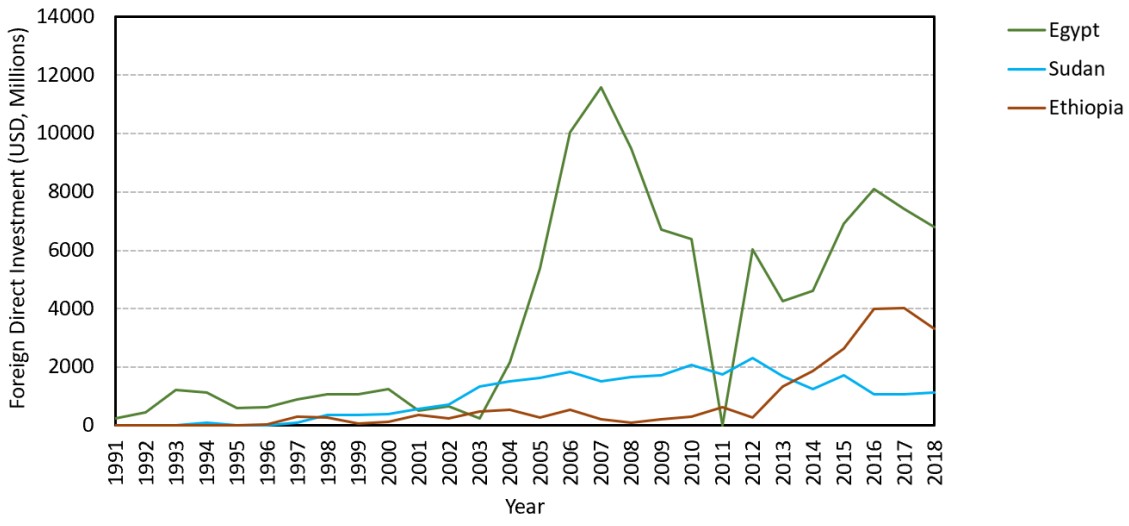

**Figure 5: The foreign direct investments from 1991 to 2018 in million USD per year (Food and Agriculture Organization Data, 2020a) in Egypt, Sudan, and Ethiopia**

The remaining part of the model inputs (food gap, energy gap) is estimated by the water resources management model of Abdelkader & Elshorbagy (2021). Runoff generation and evaporation are simulated by the SWAT (soil water assessment tool) hydrological model in a semi-distributed fashion. The model has been previously calibrated and validated by Abdelkader & Elshorbagy (2021) against daily and monthly streamflow data for the period 1981 to 2016, using the dynamically dimensioned search (DDS) calibration algorithm (Tolson & Shoemaker, 2007). The first two

380 years (i.e., 1981 to 1982) were used as a spin-up period.

The model also simulates water demand/supply for municipal, industrial, and irrigation use in the basin, as well as the food/energy production and food consumption of each riparian country. Water is supplied from each dam to different demand sites based on the priority rules from the highest priority to the lowest one: municipal, industrial, and agricultural sites. The sources of water supply for Sudan and Ethiopia are river flows and rainfall; however, due to

385 surface water scarcity, Egypt uses more diverse water sources, including deep and shallow groundwater, wastewater reuse, and desalination (Abdelkader et al. (2018)). For each agricultural, municipal, and industrial site, daily water demand is calculated. The municipal water demand is calculated by multiplying the per capita water demand by the population at a particular municipal site. According to Allen et al. (1998), the agricultural water demand is simulated by calculating crop evapotranspiration with the effect of soil moisture shortage and an irrigation efficiency factor.

Also, the industrial water demand is defined by the input data series for each site.

After simulating demand for each water demand sector, the model simulates daily hydropower generation for each dam based on dam release and a headwater reservoir. Food production is simulated for both irrigated and rainfed areas in the ENB by multiplying crop yields and agricultural areas for 20 crop groups (Abdelkader & Elshorbagy, 2021):

$$FP_{Eth,t} = A_{ig,Eth,t}.CY_{ig,Eth,t} + A_{rf,Eth,t}.CY_{rf,Eth,t}$$

$$FP_{Sud,t} = A_{ig,Sud,t}.CY_{ig,Sud,t} + A_{rf,Sud,t}.CY_{rf,Sud,t}$$

$$FP_{Egy,t} = A_{ig,Egy,t}.CY_{ig,Egy,t} \qquad\qquad (9), (10), (11)$$

where $A_{ig}\ and\ A_{rf}$ (m$^2$) indicate the actual irrigated and rainfed areas in each country. $CY_{ig}$ and $CY_{rf}$ (kg/m$^2$)

represent the corresponding crop yields in irrigated and rainfed areas. To calculate crop yields, Abdelkader & Elshorbagy (2021) simulated the adjusted crop yields due to water shortage according to Doorenbos and Kassam (1979). In addition, food consumption is simulated by multiplying per capita food consumption by the national population for each country. The water resources model does not obtain inputs from the ENB socio-hydrological model (Figure 3), so operates independently from the willingness to cooperate simulated by the ENSH model.

**3.5 Sensitivity analysis**

In order to characterize the importance of uncertainty in the model parameters, we conducted a global sensitivity analysis on the ENSH model outputs (i.e., willingness to cooperate and cooperation in the ENB). We used the Variogram Analysis of Response Surfaces (VARS) method that bridges derivative-based and variance-based

approaches (Razavi & Gupta, 2016, 2019). The average of a country's willingness to cooperate over 1983-2016 was used as the model response, with the following VARS setting: number of stars = 100; sampling resolution = 0.1 (see Razavi and Gupta (2016a) for further information on VARS sampling strategy). This setting created 21,700 model runs.

The sensitivity analysis examines 24 parameters (i.e., 9 parameters for Ethiopia and Sudan each, and 6 for Egypt)

(Table 2). All weights on the willingness to cooperate ($\alpha_1, \alpha_2, \alpha_3, \alpha_4, \beta_1, \beta_2,$ and $\beta_3$ ) vary between 0 and 1. We assumed a range of 1-20 years for the memory span and a range of 1-20 for the number of strategies.

**Table 2: Parameters varied in the sensitivity analysis and their lower and upper bounds of the Eastern Nile Basin Socio-hydrological (ENSH) model. All parameters are unitless.**

| Parameters | Description | Lower bound | Upper bound |
|---|---|---|---|
| $\alpha_1$ | A country's combined emphasis on food gap, energy gap, and future reservoir storage | 0 | 1 |
| $\alpha_2$ | A country's emphasis on political stability | 0 | 1 |
| $\alpha_3$ | A country's emphasis on foreign direct investment | 0 | 1 |
| $\alpha_4$ | A country's emphasis on memory of cooperation | 0 | 1 |
| $\beta_1$ | A country's emphasis on the food gap | 0 | 1 |
| $\beta_2$ | A country's emphasis on the energy gap | 0 | 1 |
| $\beta_3$ | A country's emphasis on future reservoir storage | 0 | 1 |
| Memory span | Number of years in which one country remembers the ENB cooperation events | 1 | 20 |
| Number of strategies | Number of ways for countries to predict the river basin cooperation status | 1 | 20 |

## 4 Results

Figure 6 shows the simulations by the ENSH model of the dynamics of the willingness to cooperate (WC) from 1983 to 2016 based on the historical basin water system (without GERD in Ethiopia). The plausibility of the simulation is
evaluated against information from the literature on the Nile River Basin. According to the model, Ethiopia's willingness to cooperate remained relatively steady in the first years, which is likely related to the Undugu project in 1983 (Kagwanja, 2007; Seide, 2014). The country's willingness to cooperate increased after 1986, which can be explained by the TECCONILE (Kagwanja, 2007; Seide, 2014). It slightly decreased in 2001 due to a high value of political stability and increased later, which is consistent with the JMP, bringing the riparian countries together in
2003 for economic improvements in the basin. After 2009, with Ethiopia's unsuccessful attempt to make Egypt return to the negotiation table, Ethiopia's willingness to cooperate decreased, due in part to large foreign direct investment. In 2010, the reduction of Ethiopia's willingness to cooperate with Sudan and Egypt is aligned with the CFA through which Ethiopia reached a cooperation agreement with the upstream countries in the absence of Sudan and Egypt (Cascão, 2009; Nicol & Cascão, 2011) and a further reduction after 2011 when the country announced the construction
of the GERD dam (Cascão & Nicol, 2016). The model suggests that the latter decrease can be explained by Egypt's weakened political stability after the Arab Spring, Ethiopia's increased political stability, higher foreign direct investment, and widening food and energy gaps. In other words, a change in political stability or foreign direct investment can change the hydro-hegemony in the basin, which is supported by Nicol & Cascão (2011).

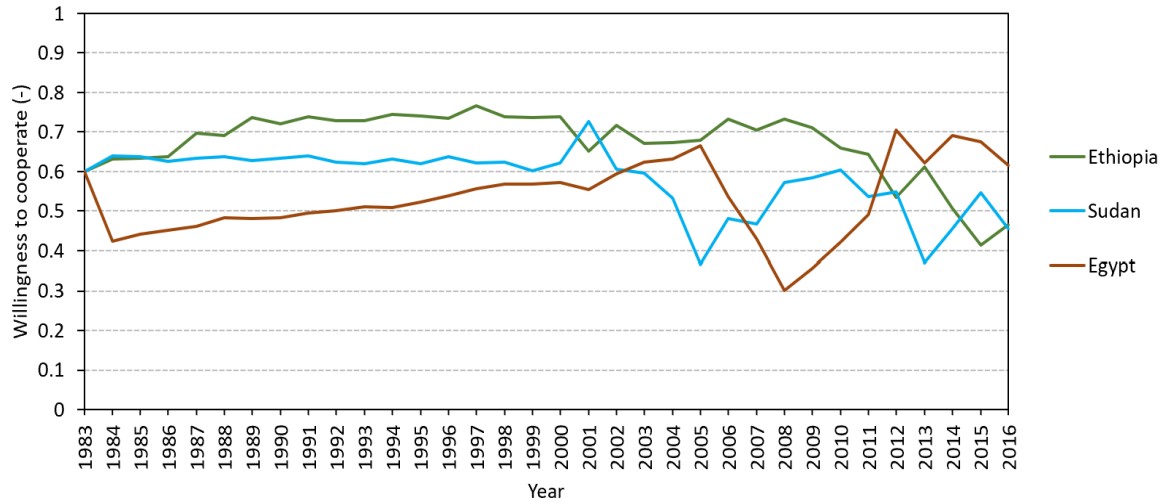

**Figure 6: Simulated riparian countries' willingness to cooperate (WC) from 1983 to 2016.**

Sudan's willingness to cooperate fluctuated around a stable value of 0.62 until 2003, except for 2001 when the willingness temporarily increased due in part to relatively low political stability. The relatively high values are related to the Undugu project and the TECCONILE (Kagwanja, 2007; Seide, 2014). After 2003, Sudan's willingness to cooperate dropped due to a perceived threat by Ethiopia to its historical water rights, before bouncing back by 2008, in line with the 2008 food crisis (Nicol & Cascão, 2011). A slight increase between 2008 and 2010 may be related to Sudan's ambition to improve its agriculture (Cascão, 2009; Nasr & Neef, 2016) and the drop between 2010 and 2013 to Sudan's reaction to CFA and construction of the GERD. After 2013, the upward trend likely reflects Sudan's position change and its support for GERD (Barnes, 2017; Swain, 2011).

Egypt's willingness to cooperate gradually rose between 1984 and 2005 in line with the food gap, to some extent political instability, and declining foreign direct investment. This upward trend can also reflect Egypt's participation in the Undugu project, TECCONILE, NBI, and JMP (Kagwanja, 2007; Seide, 2014). After 2005, Egypt's willingness dropped to its lowest value in 2008, reflecting the negotiation deadlock and Egypt's conflict with Ethiopia about Egypt's historical water rights (Nicol & Cascão, 2011). Subsequently its willingness to cooperate increased due to the decreasing political stability of Egypt during this period, eventually leading to the Arab Spring in 2011 (Knaepen & Byiers, 2017; Salman, 2013). The fluctuations in the willingness to cooperate from 2012 to 2016 can be attributed to Egypt's change of political regime, mixed reactions to GERD and a willingness to keep negotiating with the riparian countries (Tawfik, 2016).

The model estimates the basin-wide cooperation in the ENB (BC) as the average of the riparian countries' willingness to cooperate (Figure 7). The overall trend shows a slightly decreasing pattern from 1983 to 2016, indicating how mistrust and bitter memory from the past can reduce cooperation (Whittington et al., 2014). We argue that the increase in cooperation after 1984, according to the model, brought the riparian countries together for TECCONILE in 1992. Afterward, TECCONILE as guidelines for NBI, led to an increase in basin cooperation until NBI took place in 1999 (Kagwanja, 2007). These continuous agreements in the basin reveal the role of the reinforcing loop among willingness to cooperate, memory of cooperation, and cooperation in the basin. In other words, a basin agreement can act as a catalyst for further cooperation in the ENB by building trust among the three countries (Kameri-Mbote, 2007; Whittington et al., 2014).

Although the riparian countries reached these agreements, the basin cooperation decreased and dropped by 2007, reflecting the negotiation deadlock and serious conflicts among the riparian countries between 2005 and 2008 (Paisley & Henshaw, 2013). After 2008, the Arab spring in 2011 and the importance attributed by Sudan to its agricultural projects motivated the riparian countries to return to the negotiation table, which is reflected by an upward trend in Figure 7 (Cascão & Nicol, 2016). The basin cooperation went into a sharp decline in 2013 in line with the conflict

over the GERD construction (Nasr & Neef, 2016). It appears that the increased cooperation in 2014 led the riparian countries to sign a declaration on the GERD's first filling and operation in 2015 (Tawfik, 2016). However, the basin cooperation decreased shortly afterward, indicating that the declaration did not fully resolve the conflict (Cascão & Nicol, 2016).

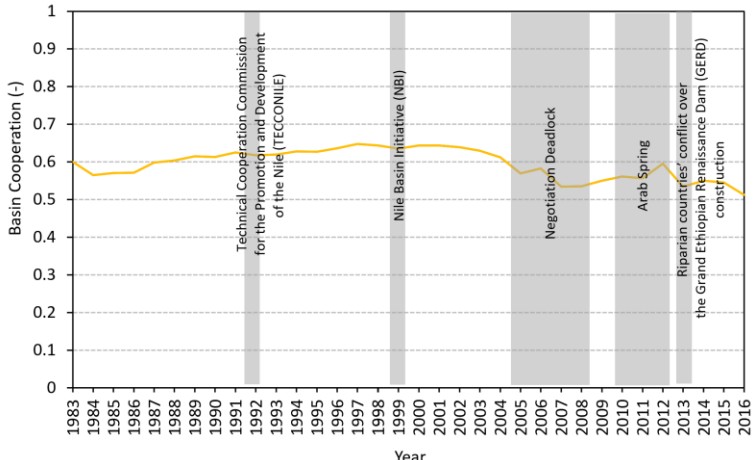

**Figure 7: Simulated cooperation in the ENB (BC) with independent information on key events from 1983 to 2016.**

Figure 8 shows the stochastic results of the ENSH model. While Figures 6-7 present one trajectory of the model outputs, Figure 8 gives the envelope of 1000 possible trajectories of how the countries emphasize each year the past basin cooperation status in their decision making (Equations 7). The results generally indicate that the uncertainty of countries' decision making based on their memory is relatively small and varies over time. The small value is likely because the countries' memory is only one part of the formulation of the countries' willingness to cooperate, and other sources of uncertainty in the model structure and the model parameters, not represented here, may further increase the uncertainty.

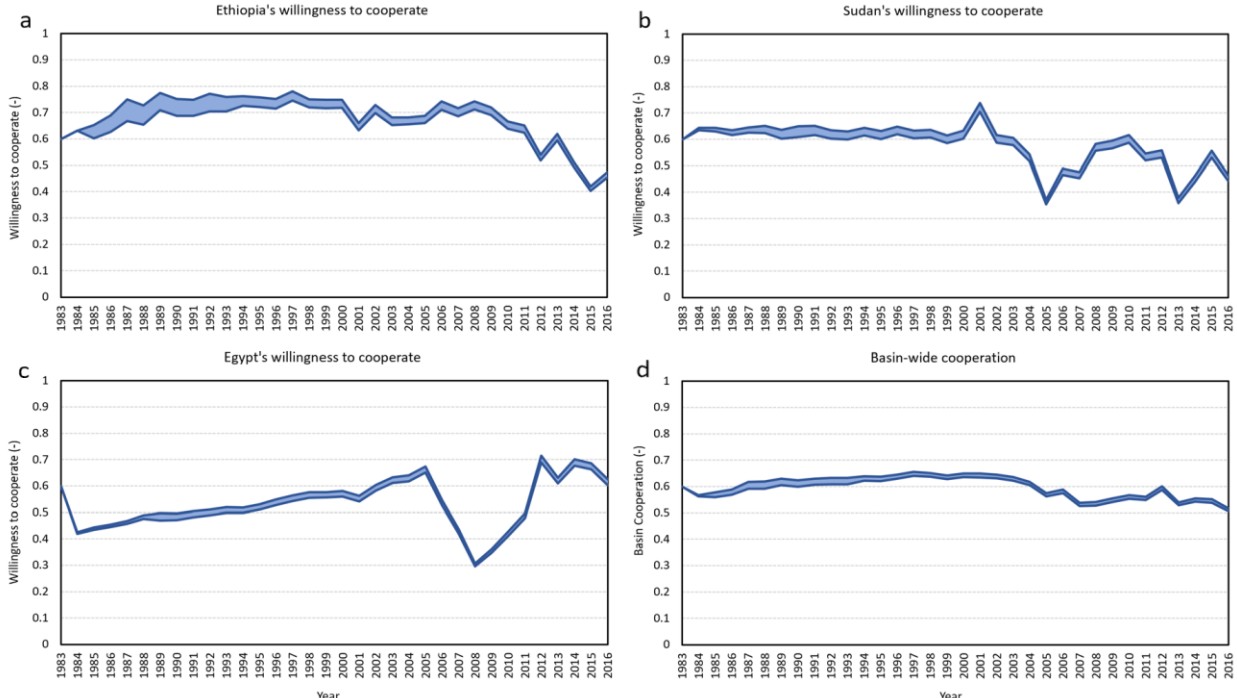

**Figure 8: (a), (b), (c) Ethiopia's, Sudan's, and Egypt's willingness to cooperate, respectively, and (d) basin-wide cooperation in the ENB with the uncertainty of the stochastic decision making (Equation 7) of the Eastern Nile Basin Socio-hydrological model indicated. The range shown is the envelope of 1000 possible trajectories. The envelope shows 1000 possible trajectories regarding the stochasticity in a country's decision making.**

Table 3 presents the results of parameter sensitivity analysis of the ENSH model. Ethiopia's weights on the energy gap ($\beta_2$ _Ethiopia) and the food gap ($\beta_1$ _Ethiopia) as well as the multiplicative variables of the food gap, the energy gap, and future reservoir storage ($\alpha_1$_Ethiopia) show the lowest rank values and are thus the most sensitive factors in Ethiopia's willingness to cooperate. This result shows that, among all socio-economic and political factors, energy gap and food gap play a more important role in Ethiopia's willingness to cooperate, which is aligned with past studies (Cascão, 2009; Nicol & Cascão, 2011). For Sudan, the weights on the energy gap ($\beta_2$ _Sudan) and food gap ($\beta_1$ _Sudan) as well as the multiplicative variables of the food gap, the energy gap and future reservoir storage ($\alpha_1$_Sudan) play an important role. However, unlike Ethiopia, Sudan's weight on the food gap ($\beta_1$ _Sudan) is more important than Sudan's weight on the energy gap ($\beta_2$ _Sudan). This result indicates that the role of agriculture sector is more important than that of energy sector in Sudan's willingness to cooperate unlike Ethiopia, which is supported by past studies (NBI, 2016; Whittington et al., 2014). For Egypt, the weights on political stability ($\alpha_2$_Egypt), memory size ($\alpha_4$_Egypt) and the foreign direct investment ($\alpha_3$_Egypt) are the most influential parameters in Egypt's willingness to cooperate. For the basin-wide cooperation in the ENB, Ethiopia's weight on the multiplicative variables of the food gap, the energy gap, and future reservoir storage ($\alpha_1$_Ethiopia) and Ethiopia's weights on the energy gap ($\beta_2$ _Ethiopia) as well as Sudan's weight on the food gap ($\beta_1$ _Sudan) play an important role. This finding implies that the basin-wide cooperation is influenced by a decrease in Ethiopia's energy gap (i.e., the construction and operation of the GERD) as well as a decrease in Sudan's food gap (i.e., agricultural development in Sudan), supported by the past studies (Abdul Latif Jameel Water and Food Systems Lab, 2014; Cascão & Nicol, 2016; Whittington et al., 2014). According to the results of parameter sensitivity analysis, we suggest that data improvements on these model parameters, e.g., by conducting surveys with the policymakers in each country, are the most relevant.

**Table 3: Parameter rankings based on the sensitivity analysis. Parameters with low values are those to which the model response is most sensitive.**

| Model parameters | Model Response | | | |
|---|---|---|---|---|
| | Parameter ranking on Ethiopia's willingness to cooperate | Parameter ranking on Sudan's willingness to cooperate | Parameter ranking on Egypt's willingness to cooperate | Parameter ranking on Basin-wide Cooperation in the ENB |
| memory span_Ethiopia | 5 | 15 | 10 | 12 |
| number of strategies_Ethiopia | 15 | 8 | 13 | 19 |
| $\alpha_1$_Ethiopia | 1 | 18 | 12 | 1 |
| $\alpha_2$_Ethiopia | 6 | 11 | 11 | 17 |
| $\alpha_3$_Ethiopia | 4 | 9 | 19 | 8 |
| $\alpha_4$_Ethiopia | 7 | 12 | 24 | 18 |
| $\beta_1$_Ethiopia | 3 | 20 | 7 | 7 |
| $\beta_2$_Ethiopia | 2 | 10 | 20 | 2 |
| $\beta_3$_Ethiopia | 18 | 22 | 15 | 22 |
| memory span_Sudan | 17 | 5 | 22 | 10 |
| number of strategies_Sudan | 24 | 19 | 5 | 23 |
| $\alpha_1$_Sudan | 9 | 1 | 14 | 5 |
| $\alpha_2$_Sudan | 19 | 4 | 16 | 11 |
| $\alpha_3$_Sudan | 14 | 7 | 21 | 16 |
| $\alpha_4$_Sudan | 23 | 6 | 17 | 15 |
| $\beta_1$_Sudan | 11 | 2 | 9 | 3 |
| $\beta_2$_Sudan | 12 | 3 | 18 | 4 |
| $\beta_3$_Sudan | 21 | 23 | 23 | 24 |
| memory span_Egypt | 16 | 14 | 2 | 9 |
| number of strategies_Egypt | 22 | 21 | 6 | 21 |
| $\alpha_1$_Egypt | 20 | 17 | 8 | 20 |
| $\alpha_2$_Egypt | 8 | 16 | 1 | 6 |
| $\alpha_3$_Egypt | 13 | 13 | 3 | 13 |
| $\alpha_4$_Egypt | 10 | 24 | 4 | 14 |

## 5 Discussion and Conclusion

When shared water resources are contested in transboundary rivers, conflict and cooperation phenomena can emerge. The Eastern Nile Basin (ENB), involving Ethiopia, Sudan, and Egypt, has a long history of conflict and cooperation. The conflict between these countries has become worse since the construction start of the Grand Ethiopian Renaissance Dam (GERD), which has drawn considerable worldwide attention. As discussed by Nicol & Cascão (2011), the existence of the GERD has changed the hydro-hegemony of the basin, and Egypt and Sudan, as previously

dominant countries in the ENB, seem to have to accept a new reality. Thus, it is important to investigate the drivers of this change in terms of riparian countries' cooperation levels to improve future cooperation in the basin. In order to explain the level of cooperation regarding water resources in the ENB, we developed a socio-hydrological model that

simulates the riparian countries' willingness to cooperate and basin-wide cooperation in the ENB from 1983 to 2016. The model results are plausible and explain how hydrological and socio-political factors can lead to the dynamics of cooperation in the basin. The findings of our model are as follows:

- Ethiopia experienced two general trends in cooperation dynamics: a relatively high willingness to cooperate between 1983 and 2009 and a subsequent decrease. The model suggests that relative political stability and foreign direct investment can explain these two different phases, along with Ethiopia's food and energy gaps. The results of sensitivity analysis also show that energy and food gaps are the most important factors in Ethiopia's willingness to cooperate. This finding suggests that improvements in Ethiopia's food gap and energy gap can be a good motivation for Ethiopia for further negotiations in the basin. Also, the model suggest that a high level of Ethiopia's relative political stability and foreign direct investment can be a barrier to further basin cooperation. This is also supported by Mason (2004) who argued that the international economic and political asymmetry can negatively affect the basin cooperation.
- Sudan's willingness to cooperate dropped between 2003 and 2008, and recovered subsequently, the latter pattern likely reflecting the 2008 food crisis in Sudan. The importance of the agricultural sector was also shown in the result of sensitivity analysis, compared to other socio-political and hydrological factors in Sudan's willingness to cooperate. This result implies that Sudan is likely to be motivated in further negotiations in the basin by improvements in its food gap.
- The drop of Egypt's willingness to cooperate around 2007 appears to be related to negotiation deadlock while, later, Egypt entered a politically unstable phase and returned to the negotiation table, which is reflected by a recovery of its willingness to cooperate. Also, the result of sensitivity analysis showed the important role of political stability and the country's memory for Egypt's willingness to cooperate. Based on Egypt's past experience, it might be challenging for Egypt to fully trust upstream countries during further negotiations because Egypt is concerned about its historical water rights of the Nile as its main water resource. Thus, we suggest that building up Egypt's trust might be the very first step for any negotiations in the basin. Such trust might be strengthened by a basic commitment by all parties that a basin-wide agreement will be the basis of infrastructure, including GERD, operation.
- At the scale of the Eastern Nile Basin, the model highlights the role of trust and good memory from the past in increasing cooperation. For example, the increase in cooperation after 1984 brought the riparian countries together for the Promotion and Development of the Nile activity (TECCONILE) in 1992. Also, the result of sensitivity analysis showed that Ethiopia's food and energy gap and Sudan's food gap are the most important factors in the basin cooperation. These findings suggest that a further cooperative agreement should be more focused on improvements in Ethiopia's food and energy gap and Sudan's food gap while assuring no significance harm to Egypt's historical water use.

The ENSH model is one of the initial attempts at quantifying and simulating the riparian countries' willingness to cooperate and basin-wide cooperation in the ENB. This study confirms the general findings of the transboundary river literature which qualitatively show that conflict and cooperation arise due to the interaction of hydrological and socio-political factors (e.g., hydropower production, food production, and political stability). We acknowledge that our results and interpretations are constrained by the model's conceptualization and assumptions. The ENSH model is built on the existing knowledge of the processes, but future studies could use alternative social theories and hypotheses, thereby improving our perceptual understanding of how the system works. In this regard, we should point out that our primary modeling purpose has been 'diagnostic learning' by simulating the past, complex behaviours pertaining to the conflict and cooperation in ENB – see the discussion in Razavi et al. (2022) for models as a tool for diagnostic learning. We believe this work, along with previous work cited in this paper, constitute the first steps towards building a predictive model for such phenomena to be used for future decision support.

Based on the results of the parameter sensitivity analysis, future studies should improve information on Ethiopia and Sudan's weights on the energy and food gap as well as Egypt's weights on political stability and memory span. Another potential source of uncertainty of the ENSH model is the input data (i.e., energy production, food consumption, food production, foreign direct investment, future reservoir storage, potential energy capacity, and relative political stability). The significance of such uncertainty in inputs on the outcome may be assessed through sensitivity analysis in future work. The ENSH model could also be coupled with a water resources model to explore

the evolution of cooperation pathways among the riparian countries affected by the GERD operation, which can assist in identifying actions that may fuel conflict in the basin.

## Acknowledgment

The authors acknowledge the support from the 2019 Summer Institute on Sociohydrology and Transboundary Rivers at Yunnan University, China. We would like to especially thank Drs. Marlies Barendrecht, Elena Mondino, and Handriyanti Diah Puspitarini for their help with an initial conceptualization of this work. This research is funded by the Integrated Modeling Program for Canada and the Ph.D. Excellence Scholarship from the School of Environment and Sustainability at the University of Saskatchewan as well as Razavi's Discovery Grant from the Natural Sciences and Engineering Research Council of Canada.

## Code and data availability

The data are available on request from the corresponding author (mohammad.ghoreishi@usask.ca).

## Author contributions

MG and AE developed the model conceptualization with the contribution of GB and MS. MG coded and ran the model and processed the results. AA contributed to the water resources component of the model. MG, AE, SR, GB, and MS contributed to the analysis and discussion of the results. MG wrote the initial draft and all co-authors reviewed and edited the paper.

## Competing interests

The authors declare that they have no conflict of interest.

595

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
