# Peer review of "Cooperation in a Transboundary River Basin: a Large Scale Socio-hydrological Model of the Eastern Nile"

_Hydrology and Earth System Sciences, 2022_

## Author Comment (AC1)

We appreciate the reviewers' time and effort in reviewing our paper. We believe that these comments will improve the paper. This document contains copies of all the comments of the Reviewers (in blue text) and our responses to them (in black text).

**Response to Reviewer 1:**

R1.1. While I find the overall paper topic to be interesting and important, it seems the emphasis on being one of the first to perform a quantitative basis of water boundary conflicts is far-fetched. A quick search reveals several papers that have proposed the same end-goal, yet were not cited. How does this study extend, contrast, confirm, or completely refute such previous studies? To name a few:

- Avisse, N., Tilmant, A., Rosenberg, D., & Talozi, S. (2020). Quantitative assessment of contested water uses and management in the conflict-torn Yarmouk River Basin. Journal of Water Resources Planning and Management, 146(7), 05020010.
- Jacob-Rousseau, N. (2015). Water diversions, environmental impacts and social conflicts: the contribution of quantitative archives to the history of hydraulics. French cases (nineteenth century). Water History, 7(1), 101-129.
- Beck, L., Bernauer, T., Siegfried, T., & Böhmelt, T. (2014). Implications of hydro-political dependency for international water cooperation and conflict: Insights from new data. Political Geography, 42, 23-33.
- Van Baalen, S., & Mobjörk, M. (2018). Climate change and violent conflict in East Africa: Integrating qualitative and quantitative research to probe the mechanisms. International Studies Review, 20(4), 547-575.
- Kilgour, D. M., & Dinar, A. (2001). Flexible water sharing within an international river basin. Environmental and Resource Economics, 18(1), 43-60.
- Tinti, A. (2015). Water scarcity and regional fragmentation in the Middle East: A quantitative assessment. Politikon: The IAPSS Journal of Political Science, 27, 177-205.
- Madani, K. (2010). Game theory and water resources. Journal of Hydrology, 381(3-4), 225-238.
- Grech-Madin, C., Döring, S., Kim, K., & Swain, A. (2018). Negotiating water across levels: A peace and conflict "Toolbox" for water diplomacy. Journal of Hydrology, 559, 100-109.
- + many others

If the overall paper's contribution is to be a premier study emphasizing quantitative components of water conflict issues, then a deeper literature review and framing within the existing body of research is essential. If the overall paper's contribution is something else, consider changing the abstract to emphasize that component.

We regret that the novelty of this study is not clearly stated in the manuscript. To address this issue, we will improve the literature review based on your suggested papers and emphasize the contribution of this study.

While we acknowledge all previous studies in conflict and cooperation, including those you mentioned, we believe that quantifying the dynamics of cooperation with both socio-political and hydrological factors has been elusive in the literature of conflict and cooperation studies. In fact, the contribution of this study is to quantify this phenomenon over time after investigating the important socio-political and hydrological factors in the Eastern Nile River Basin using causal feedback.

R1.2. General: There are quite a bit of acronyms used in this paper, which is fine, but it might be helpful to the reader to include a list of all acronyms at the forefront or as an Appendix to the paper.

We will add a list of all acronyms in the appendix of the revised manuscript.

R1.3. General: It was not immediately clear at first read why the ENB was emphasized for conflict out of the entire Nile – do the other countries not have qualms over the water usage? A quick search suggests that many of the countries along the Nile have had conflict to-date over water. (e.g., https://www.tandfonline.com/doi/pdf/10.1080/17531050701625565). For example, even though perhaps Ethiopia and Sudan are most vocal about the Nile dam, such decisions significantly impact Kenyans and Ugandans. It is acceptable to limit the scope of the study to a portion of such a large river basin, but I was just unclear as to the rationale at first read of the paper.

Thank you for raising this point. According to lines 47-48, "Water conflicts are more severe in the Eastern Nile Basin (ENB), where Ethiopia, Sudan, and Egypt are located, due to water scarcity." This conflict has been aggravated over the past several years due to the GERD construction in Ethiopia. The conflict among the three countries has been attracting significant international attention, and thus, the scope of our manuscript is well justified. This is why we focused on the ENB in this study. We will emphasize this issue further to avoid any confusion. However, we will also highlight that solving the ENB conflict does not mean addressing the entire Nile Basin issue.

R1.4. Fig. 1 is good, but a few minor suggestions: Try to avoid using pink and red to differentiate very similar boundary types (e.g., use a more contrasting color); considering adding the datum to the caption for referencing the lat/lon values (I'm sure it's the standard WGS 1984 datum, but it always helps to include this type of information in GIS-based maps); consider adding to the legend what the dark blue polygon boundary represents; ensure final figure to the Journal (usually in PDF format) is very high-resolution, as it appears blurry in the current embedded format.

In the revised paper, we will improve the quality of Figure 1, as you pointed out.

R1.5. Introduction (General): Overall introduction appears to have all of the "pieces" there but is put together in a manner that reads as disjointed in thought. A bit more effort is encouraged in telling the story here, by starting with the general problem, its importance, and then narrowing down into what has been done thus far to address, how those still have gaps, and then finally what this paper brings to the table.

We will use your suggested outline to improve the Introduction Section.

R1.6. Line 30-33: What is C&C? In general, I think there needs to be a bit more elaboration on the overall "big picture problem" and why it is important to the reader (and society) before jumping into details of the literature, particularly with acronyms that are not explicit.

Thank you for raising this issue. C&C stands for conflict and cooperation. We will address this issue in the revision.

R1.7. Line 33-35: Sentence seems to belong before explanation of literature. Describe water conflicts, political agreements, and the overall mindset of socio-hydrology before delving into details.

We will improve the introduction section by adding the literature review of socio-hydrology and a description of general terms in transboundary rivers (e.g., water conflict, political agreements).

R1.8. Line 38: Describe hydro-hegemony. Remember, HESS is read by a broad group of hydrologists and earth scientists who may not be familiar with the common lingo in socio-hydrology.

In this study, hydro-hegemony is the leadership or dominance of a riparian country over other riparian countries in a transboundary river basin due to a riparian country's upstream position or historical water rights. We will add this definition in the revised manuscript.

R1.9. Line 57-60: While this literature on the Nile is robust, it is too focused on the geographical case study. Further literature, which is important for framing the overall novelty of the paper, is generally missing.

According to our answer R1.1, we will clarify the contribution of this study in the revision and tie it to the available literature.

R1.10. Line 70: Cooperation "and conflict"?

Yes, it is a typo error. We will fix it in the revision.

R1.11. Line 70: Choice of word "confronted" here seems out of place.

The word will be replaced with "validated" in the revision.

R1.12. Lines 83-133: This large amount of text could be significantly condensed and/or added to SI. It is not yet clear how this historically based narrative can be considered quantitative (perhaps semi-quantitative or fuzzy-based transformation from qualitative to semi-quantitative?).

These lines initially suggested the most important socio-political and hydrological factors in conflict and cooperation dynamics in the ENB in a qualitative manner. These factors were taken into account in the following section (3.1 variables) where we identified all these factors and

validated their significant role in ENB's cooperation dynamics with the local and general studies in conflict and cooperation dynamics. At the end of this section (3.1), Table 1 showed how we obtained the quantitative data of all these factors from different sources. We believe it is important to present these factors and how we reached at them because their identification is important for this study.

R1.13. 3 Method: Temporally, what data? Precipitation, streamflow, where did you get it, how was it verified or roughly calibrated? What are the "units" being discussed? What is the basis of this model? If it is meant to be a stylized socio-hydrological model, this should be discussed somewhere prior to identifying the system variables and assumptions.

All model variables are shown in Table 1 with their sources of data. In section 3.4, we divided the model inputs into two categories: the inputs taken from the literature and the inputs taken from a water resources model. We explained all data sources taken from the literature and their time span. We also explained how the water resources model, which was developed and validated by Abdelkader & Elshorbagy (2021), estimated the remaining part of our model inputs (i.e., their equations were shown). In the revised manuscript, we will provide their units. As it is explained in section 3.4, one part of our data (food gap and energy gap) is based on the water resources management model of Abdelkader & Elshorbagy (2021), which was previously calibrated and validated. The rest of the data are based on the Food and Agriculture Organization (FAO) data and Nile River Basin reports. Although we acknowledge the limitations of our developed model, we do not consider it as a stylized socio-hydrological model. This is because our model is based on a previously calibrated and validated model (Abdelkader & Elshorbagy, 2021), and we also validated our model output with independent qualitative data from the literature in the result section.

R1.14. Overall Methodological section is in piece-meal nature and is not, in my opinion, at a quality and coherency level for publication. I do think there are strong bases here, and a well-intended study was conducted, but the way it is written and presented could use further explanation for the reader to be introduced to this type of model-thinking.

We regret that the current method section is not as clear and coherent as we thought. We will strive to improve the readability of this Section without increasing the paper length significantly, as it is already lengthy.

R1.5. Figure 3: A suggestion – consider changing the Eth, Sud, Egy nomenclature to be represented by different color nodes, rather than text additions.

We will use different colors to represent each country in the revised manuscript.

R1.16. Section 3.3: I do not agree that this is a causal "loop" diagram. The loops, in terms of reinforcing/balancing and how they then interact dynamically amongst one another are not depicted graphically. Rather, this is a causal feedback diagram. There are 3 loops in the middle, but they are all reinforcing, which would not make much sense in terms of figuring out overall causality as the entire system would keep, theoretically, reinforcing itself on a forever trajectory.

We will change it to "the causal feedback diagram". Regarding the reinforcing loop, as you mention, a reinforcing loop by itself leads to a forever trajectory. However, these reinforcing loops in our hypothesized system only show a part of our system, and they are explicitly surrounded by complex feedback. For example, water supply for hydropower can increase through an increase in the energy gap. This increase in water supply for hydropower increases energy production and thus reducing energy gap (i.e., balancing loop).

R1.17. Eq. 1-8: It is hard to review the equations, when the overall picture and causal depiction is unclear. Perhaps a bit more explanation on these variables, and/or why they were hypothesized as such, and/or tables showing the variables and their dynamic simulation in the SI would help the reader follow?

In the revised manuscript, we will improve the description of the overall picture of the work. We will also add a table including all equations and variables in the supporting information.

R1.18. 4 Results: The explanations here are helpful in making sense of the previous section. I still recommend a deeper review of how the work is being presented and organized for overall readability.

As confirmed in a previous comment (R1.1), we will improve the introduction to address this issue.

R1.19. General Note: All equation variables, when listed in-text, are showing up as very large, blurry, and distorted in the PrePrint PDF. I am not sure if this is an issue with the HESS conversion format into PDF, but please verify that the texts are provided in the proper fonts.

We will address this issue in the revised manuscript.

R1.20. Conclusion: Most of the conclusion section is actually a further listing of detailed Results. Please consider re-phrasing the Results section to be cohesive in one read, and then in the Conclusions, highlight the overall take-away, not re-listing the methodological outputs.

In the revised manuscript, we will revise the Conclusion Section to highlight the main take-home message and we will try to avoid any listing of results.

---

## Author Comment (AC2)

We appreciate the reviewers' time and effort in reviewing our paper. We believe that these comments will improve the paper. This document contains copies of all the comments of the Reviewers (in blue text) and our responses to them (in black text).

**Response to Reviewer 2:**

This paper investigated the main factors in the riparian countries' willingness to cooperate in the Eastern Nile River Basin, involving Ethiopia, Sudan, and Egypt, from 1983 to 2016. A quantitative model of the willingness to cooperate at the national and river basin scales. was developed. It was found that relative political stability and foreign direct investment can explain Ethiopia's decreasing willingness to cooperate between 2009 and 2016.

Several key points for improvement of this manuscript:

R2.1. Since 2012 even earlier, the socio-hydrological community has developed a lot of socio-hydrological models to simulate various emergent human-water phenomena. Recently, on the same issue, the socio-hydrological models on conflict and cooperation either conceptually or empirically have been published. I do not understand the authors can ignore them. This leads to another concern: What is New in this manuscript?

Thank you for raising this issue. In the revised manuscript, we will strengthen the literature review on socio-hydrology, including those published regarding the conflict and cooperation phenomenon (e.g., Lu et al., (2021)). According to our response to the first reviewer (R1.1), we believe that quantifying the dynamics of cooperation value with both socio-political and hydrological factors has been elusive in the literature of conflict and cooperation studies, including socio-hydrological research. In fact, the contribution of this study is to quantify riparian countries' willingness to cooperate as well as basin cooperation over time after investigating the important socio-political and hydrological factors in the Eastern Nile River Basin using causal feedback.

R2.2. The conceptualization model should appear in the beginning of Section Methods. Otherwise, the readers could not understand what you talked about.

We regret that the current method section has led to confusion. We do believe that we need to identify and introduce all variables before explaining their causal relationship. However, to address your valid concern, we will add a description at the beginning of the method section to explain the big picture of this study and the overall methodology.

R2.4. More theoretical evidence should be given to justify your selection on those societal variables.

Lines 83-133 suggested the most important socio-political and hydrological factors in conflict and cooperation dynamics in the Eastern Nile Basin (ENB) in a qualitative manner. These factors were taken into account in the following section (3.1 variables). Section 3.1 provides a detailed

explanation of each social variable with theoretical evidence. This selection was supported by both local studies and general literature of transboundary rivers. However, we will add more references to address your concern.

R2.5. Should have more direction discussion/comparison on your simulated C&C and those from independent sources. I am shocked by the manuscript does not have a discussion/implication section. After you present your results. SO WHAT?

The result section provides an explanation of the dynamics of cooperation in the ENB, supported by local studies in the basin. In fact, each country's simulated willingness to cooperate was compared to independent qualitative sources. Also, there are some implications in the middle of the discussion (e.g., lines 471-473 provide the implication of the sensitivity analysis). However, to address your concern, we will strive to add more implications in the section without increasing the paper length significantly, as it is already lengthy. We regret that the take-home message is not as clear as we thought. We will also improve the conclusion section to clarify the take-home message.

---

## Author Response (AR1)

We appreciate the reviewers' time and effort in reviewing our paper. We believe that these comments helped us improve the paper. This document contains copies of all the comments of the Reviewers (in blue text) and our responses to them (in black text).

**Response to Reviewer 1:**

R1.1. While I find the overall paper topic to be interesting and important, it seems the emphasis on being one of the first to perform a quantitative basis of water boundary conflicts is far-fetched. A quick search reveals several papers that have proposed the same end-goal, yet were not cited. How does this study extend, contrast, confirm, or completely refute such previous studies? To name a few:

- Avisse, N., Tilmant, A., Rosenberg, D., & Talozi, S. (2020). Quantitative assessment of contested water uses and management in the conflict-torn Yarmouk River Basin. Journal of Water Resources Planning and Management, 146(7), 05020010.
- Jacob-Rousseau, N. (2015). Water diversions, environmental impacts and social conflicts: the contribution of quantitative archives to the history of hydraulics. French cases (nineteenth century). Water History, 7(1), 101-129.
- Beck, L., Bernauer, T., Siegfried, T., & Böhmelt, T. (2014). Implications of hydro-political dependency for international water cooperation and conflict: Insights from new data. Political Geography, 42, 23-33.
- Van Baalen, S., & Mobjörk, M. (2018). Climate change and violent conflict in East Africa: Integrating qualitative and quantitative research to probe the mechanisms. International Studies Review, 20(4), 547-575.
- Kilgour, D. M., & Dinar, A. (2001). Flexible water sharing within an international river basin. Environmental and Resource Economics, 18(1), 43-60.
- Tinti, A. (2015). Water scarcity and regional fragmentation in the Middle East: A quantitative assessment. Politikon: The IAPSS Journal of Political Science, 27, 177-205.
- Madani, K. (2010). Game theory and water resources. Journal of Hydrology, 381(3-4), 225-238.
- Grech-Madin, C., Döring, S., Kim, K., & Swain, A. (2018). Negotiating water across levels: A peace and conflict "Toolbox" for water diplomacy. Journal of Hydrology, 559, 100-109.
- + many others

If the overall paper's contribution is to be a premier study emphasizing quantitative components of water conflict issues, then a deeper literature review and framing within the existing body of research is essential. If the overall paper's contribution is something else, consider changing the abstract to emphasize that component.

While we acknowledge all previous studies in conflict and cooperation, including those you mentioned, we believe that quantifying the dynamics of cooperation with both socio-political and hydrological factors has been elusive in the literature of conflict and cooperation studies. In fact, the contribution of this study is to quantify this phenomenon over time after investigating the

important socio-political and hydrological factors in the Eastern Nile River Basin using a causal feedback framework (Lines 86-91). To address your concern, we have improved the literature review based on your suggested papers and emphasized the contribution of this study (Lines 46-63):

*"The transboundary rivers have been receiving significant attention by many studies (e.g., Elhance, 1999; Kilgour & Dinar, 2001; Wolf, 2007). The literature of transboundary rivers has generally focused on pathways towards resolving conflicts (e.g., Madani et al., 2014; Rogers, 1969; Zarezadeh et al., 2012), analyzing conflict and cooperation (C&C;e.g., Mirumachi & Van Wyk, 2010; Wolf, 2007; Wolf et al., 2003), and investigating influential factors in C&C (e.g., Dinar et al., 2010; Zeitoun et al., 2011), often in a scenario-based context. Recently, C&C in transboundary water systems has attracted the attention of socio-hydrological research, which focuses on the coevolutionary behavior between social and hydrological systems (Sivapalan et al., 2012). The endogeneity of humans in water systems has been the subject of numerous socio-hydrological studies using a variety of methods, including those incorporating socio-economic drivers (Aghaie et al., 2020; Elshafei et al., 2014) or those employing concepts of social memory (Di Baldassarre et al., 2013; Gonzales & Ajami, 2017a) and collective behaviors (Du et al., 2017a; Garcia et al., 2016b). Compared to other studies on transboundary rivers, socio-hydrological research emphasizes quantifying C&C dynamics by including both socio-political and hydrological factors in modeling (e.g., Lu et al., 2021), and providing a general framework on C&C (e.g., Wei et al., 2022). However, previous studies on socio-hydrology in transboundary rivers did not focus on the important concept of social memory and quantitative components of C&C phenomena (e.g., political stability). Also, the advantage of qualitative data and narratives for model validation is elusive in the current socio-hydrological research on transboundary rivers. Thus, more research needs to be done to understand C&C in other transboundary rivers and investigate the associated socio-political factors with the use of qualitative data for model validation. This study is intended to contribute to filling some of these research gaps."*

R1.2. General: There are quite a bit of acronyms used in this paper, which is fine, but it might be helpful to the reader to include a list of all acronyms at the forefront or as an Appendix to the paper.

We have added a list of all acronyms in the appendix of the revised manuscript (Table A2).

R1.3. General: It was not immediately clear at first read why the ENB was emphasized for conflict out of the entire Nile – do the other countries not have qualms over the water usage? A quick search suggests that many of the countries along the Nile have had conflict to-date over water. (e.g., https://www.tandfonline.com/doi/pdf/10.1080/17531050701625565). For example, even though perhaps Ethiopia and Sudan are most vocal about the Nile dam, such decisions significantly impact Kenyans and Ugandans. It is acceptable to limit the scope of the study to a portion of such a large river basin, but I was just unclear as to the rationale at first read of the paper.

Thank you for raising this point. According to lines 65-66, "*Water conflicts are more severe in the Eastern Nile Basin (ENB), where Ethiopia, Sudan, and Egypt are located, due to water scarcity.*" To address this issue, we have added the following lines (lines 73-76):

*"The conflict over the GERD has been aggravated over the past several years. Importantly, the conflict among the three countries has been attracting significant international attention, and therefore, we focused on the ENB in this study, which can contribute to addressing larger issues across the entire Nile Basin."*

R1.4. Fig. 1 is good, but a few minor suggestions: Try to avoid using pink and red to differentiate very similar boundary types (e.g., use a more contrasting color); considering adding the datum to the caption for referencing the lat/lon values (I'm sure it's the standard WGS 1984 datum, but it always helps to include this type of information in GIS-based maps); consider adding to the legend what the dark blue polygon boundary represents; ensure final figure to the Journal (usually in PDF format) is very high-resolution, as it appears blurry in the current embedded format.

In the revised paper, we have improved the quality of Figure 1, as you pointed out.

R1.5. Introduction (General): Overall introduction appears to have all of the "pieces" there but is put together in a manner that reads as disjointed in thought. A bit more effort is encouraged in telling the story here, by starting with the general problem, its importance, and then narrowing down into what has been done thus far to address, how those still have gaps, and then finally what this paper brings to the table.

To address your concern, we have reorganized the paragraphs in the Introduction and highlighted the gaps and the state-of-the-art (Lines 26-91).

R1.6. Line 30-33: What is C&C? In general, I think there needs to be a bit more elaboration on the overall "big picture problem" and why it is important to the reader (and society) before jumping into details of the literature, particularly with acronyms that are not explicit.

Thank you for raising this issue. C&C stands for conflict and cooperation, which has been corrected in the manuscript. As mentioned in R1.5, we have modified the introduction section.

R1.7. Line 33-35: Sentence seems to belong before explanation of literature. Describe water conflicts, political agreements, and the overall mindset of socio-hydrology before delving into details.

As mentioned in R1.5, we have modified the introduction section. Also, we have added the following sentence to address your concern (lines 26-29)*:*

*"The contested use of these shared water resources can lead to water conflicts (i.e., a dispute between countries over the rights to water resources) or cooperative agreements (i.e., the peaceful management and use of water resources among countries) (Wolf et al., 2003; Zeitoun & Mirumachi, 2008)."*

R1.8. Line 38: Describe hydro-hegemony. Remember, HESS is read by a broad group of hydrologists and earth scientists who may not be familiar with the common lingo in socio-hydrology.

To address this issue, we have added the following (lines 38-40):

*"In this study, hydro-hegemony is defined as the leadership or dominance of a riparian country over other riparian countries in a transboundary river basin due to a riparian country's upstream position or historical water rights."*

R1.9. Line 57-60: While this literature on the Nile is robust, it is too focused on the geographical case study. Further literature, which is important for framing the overall novelty of the paper, is generally missing.

According to our answer R1.1, we have clarified the contribution of this study in the revision and tie it to the available literature.

R1.10. Line 70: Cooperation "and conflict"?

Yes, it is a typo error. We have fixed it in the revision.

R1.11. Line 70: Choice of word "confronted" here seems out of place.

The word has been replaced with "validated" in the revision.

R1.12. Lines 83-133: This large amount of text could be significantly condensed and/or added to SI. It is not yet clear how this historically based narrative can be considered quantitative (perhaps semi-quantitative or fuzzy-based transformation from qualitative to semi-quantitative?).

These lines initially suggested the most important socio-political and hydrological factors in conflict and cooperation dynamics in the ENB in a qualitative manner. These factors were taken into account in the following section (3.1 variables) where we identified all these factors and validated their significant role in ENB's cooperation dynamics with local and general studies in conflict and cooperation dynamics. At the end of this section (3.1), Table 1 showed how we obtained the quantitative data of all these factors from different sources. We believe it is important to present these factors and how we reached at them because their identification is an important piece of this study.

R1.13. 3 Method: Temporally, what data? Precipitation, streamflow, where did you get it, how was it verified or roughly calibrated? What are the "units" being discussed? What is the basis of this model? If it is meant to be a stylized socio-hydrological model, this should be discussed somewhere prior to identifying the system variables and assumptions.

All model variables are shown in Table 1 with their sources of data. In section 3.4, we divided the model inputs into two categories: the inputs taken from the literature and the inputs taken from a water resources model. We explained all data sources taken from the literature and their time span. We also explained how the water resources model, which was developed and validated by Abdelkader & Elshorbagy (2021), estimated the remaining part of our model inputs (i.e., their equations were shown). We have provided the units in the body of the manuscript and in Table 1, as well as Appendix (Table A3). As it is explained in section 3.4, one part of our data

(food gap and energy gap) is based on the water resources management model of Abdelkader & Elshorbagy (2021), which was previously calibrated and validated. The rest of the data are based on the Food and Agriculture Organization (FAO) data and Nile River Basin reports. Although we acknowledge the limitations of our developed model, we do not consider it as a stylized socio-hydrological model. This is because our model is based on a previously calibrated model (Abdelkader & Elshorbagy, 2021), and we also validated our model output with independent qualitative data from the literature in the Results section.

R1.14. Overall Methodological section is in piece-meal nature and is not, in my opinion, at a quality and coherency level for publication. I do think there are strong bases here, and a well-intended study was conducted, but the way it is written and presented could use further explanation for the reader to be introduced to this type of model-thinking.

To address your concern, we have added a few lines in the Methods (lines 278-283) and also added a table (Table A3) to summarize all equations for better readability:

*"We set up the equations of willingness to cooperate based on human behavior in resource dilemmas. Consistent findings show that decision behaviors in resource dilemmas are mostly individualism (i.e., the drive to prioritize one's own interests) and competition (the drive to increase relative gains, the gap between one's achievement and that of the other) (Brewer & Kramer, 1986; Parks & Vu, 1994; Roch & Samuelson, 1997). Thus, here, countries' willingness to cooperate is evaluated based on their relative socio-economic gains (Kopelman et al., 2002; Mason, 2004)."*

R1.5. Figure 3: A suggestion – consider changing the Eth, Sud, Egy nomenclature to be represented by different color nodes, rather than text additions.

We have used different colors to represent each country in the revised manuscript.

R1.16. Section 3.3: I do not agree that this is a causal "loop" diagram. The loops, in terms of reinforcing/balancing and how they then interact dynamically amongst one another are not depicted graphically. Rather, this is a causal feedback diagram. There are 3 loops in the middle, but they are all reinforcing, which would not make much sense in terms of figuring out overall causality as the entire system would keep, theoretically, reinforcing itself on a forever trajectory.

We have changed it to "the causal feedback diagram". Regarding the reinforcing loop, as you mention, a reinforcing loop by itself leads to a forever growth trajectory. However, these reinforcing loops in our hypothesized system only show a part of our system, and they are explicitly surrounded by complex feedback. For example, water supply for hydropower can increase through an increase in the energy gap. This increase in water supply for hydropower increases energy production and thus reducing energy gap, which reduces the need for further power generation (i.e., balancing loop).

R1.17. Eq. 1-8: It is hard to review the equations, when the overall picture and causal depiction is unclear. Perhaps a bit more explanation on these variables, and/or why they were hypothesized

as such, and/or tables showing the variables and their dynamic simulation in the SI would help the reader follow?

According to R1.14, To address your concern, we have added a few lines in the Methods (lines 278-283) and also added a table (Table A3) to summarize all equations for better readability.

R1.18. 4 Results: The explanations here are helpful in making sense of the previous section. I still recommend a deeper review of how the work is being presented and organized for overall readability.

As confirmed in a previous comment (R1.1), we have improved the introduction to address this issue.

R1.19. General Note: All equation variables, when listed in-text, are showing up as very large, blurry, and distorted in the PrePrint PDF. I am not sure if this is an issue with the HESS conversion format into PDF, but please verify that the texts are provided in the proper fonts.

We have addressed this issue.

R1.20. Conclusion: Most of the conclusion section is actually a further listing of detailed Results. Please consider re-phrasing the Results section to be cohesive in one read, and then in the Conclusions, highlight the overall take-away, not re-listing the methodological outputs.

In the revised manuscript, we have changed the Conclusion section to the Discussion and Conclusion section to discuss the results and clarify the take-home message (lines 502-557).

**Response to Reviewer 2:**

This paper investigated the main factors in the riparian countries' willingness to cooperate in the Eastern Nile River Basin, involving Ethiopia, Sudan, and Egypt, from 1983 to 2016. A quantitative model of the willingness to cooperate at the national and river basin scales. was developed. It was found that relative political stability and foreign direct investment can explain Ethiopia's decreasing willingness to cooperate between 2009 and 2016.

Several key points for improvement of this manuscript:

R2.1. Since 2012 even earlier, the socio-hydrological community has developed a lot of socio-hydrological models to simulate various emergent human-water phenomena. Recently, on the same issue, the socio-hydrological models on conflict and cooperation either conceptually or empirically have been published. I do not understand the authors can ignore them. This leads to another concern: What is New in this manuscript?

Thank you for raising this issue. As mentioned earlier in regard to R1.1 and R1.5, in the revised manuscript, we have strengthened the literature review on socio-hydrology, including those published regarding the conflict and cooperation phenomenon (e.g., Lu et al., (2021)). According to our response to the first reviewer (R1.1), we believe that quantifying the dynamics of cooperation value with both socio-political and hydrological factors has been elusive in the literature of conflict and cooperation studies, including socio-hydrological research. In fact, the contribution of this study is to quantify riparian countries' willingness to cooperate as well as basin cooperation over time after identifying the important socio-political and hydrological factors in the Eastern Nile River Basin using a causal feedback framework.

R2.2. The conceptualization model should appear in the beginning of Section Methods. Otherwise, the readers could not understand what you talked about.

We regret that the current Methods section has led to confusion. We believe that we need to identify and introduce all variables before explaining their causal relationship. However, we have added a description at the beginning of the Methods section to explain the big picture of this study and the overall methodology (lines 156-163):

*"In section 2, we initially detected the most important socio-political and hydrological factors in C&C dynamics in the ENB through the existing narrative. These factors are taken into account in section 3.1 where we fully identify all these factors and validate their significant role in ENB's C&C dynamics with local and general studies in transboundary rivers. In section 3.2, we conceptualize the interactions of the C&C dynamics in the basin based on the ENB literature. Spatially, each of Ethiopia, Sudan, and Egypt is considered as one unit, and temporally we adopt an annual resolution. In section 3.3, we introduce the hypothesized equations to simulate the riparian countries' willingness to cooperate and the basin-wide cooperation. All model inputs and the setting of sensitivity analysis are introduced in sections 3.4 and 3.5, respectively."*

R2.4. More theoretical evidence should be given to justify your selection on those societal variables.

Lines 94-153 suggested the most important socio-political and hydrological factors in conflict and cooperation dynamics in the Eastern Nile Basin (ENB) in a qualitative manner. These factors were taken into account in the following section (3.1 variables). Section 3.1 provides a detailed explanation of each social variable with theoretical evidence. This selection was supported by both local studies and general literature of transboundary rivers. We have added a few lines to better clarify this issue, as mentioned regarding R2.2.

R2.5. Should have more direction discussion/comparison on your simulated C&C and those from independent sources. I am shocked by the manuscript does not have a discussion/implication section. After you present your results. SO WHAT?

The result section provides an explanation of the dynamics of cooperation in the ENB, supported by local studies in the basin. In fact, each country's simulated willingness to cooperate was compared to independent qualitative sources. Also, there are some implications in the middle of the discussion (e.g., lines 495-496 provide the implications of the sensitivity analysis). However, to address your concern, we have changed the Conclusion section to the Discussion and Conclusion section to discuss the results and clarify the take-home message (lines 502-557).

---

## Author Response (AR2)

**Response to Reviewers' Comments**

**Associate Editor:**

Based on my own reading, the paper provides an improved representation of key processes in the sociohydrological modelling of transboundary water conflict and cooperation (C&C) in the context of previous efforts. It also provides new insights to understand the C&C dynamics occurring in the East Nile Basin. However, as expressed by the Referees, the major concern about the paper's current form is the vague expression of its novelty. Please check the Referee's comments for more detail. I agree with the concern although I acknowledge the paper does improve our knowledge of the C&C modelling and our understanding of the real Nile water C&C situation. In my understanding, the modelling work of this study is built on the previous method proposed by Lu et al. (2021) and the framework proposed by Wei et al. (2022) in this issue. It can be better understood if the authors express in a way: how your model is improved from Lu's work and how the model complies with Wei et al. framework or improves it. The paper can also highlight the insights gained from the modelling for C&C understanding in the Nile Basin.

I would like to ask the authors to revise the manuscript based on all the comments and I will make a further review. Thanks for your efforts.

Thank you for your time to review our revised manuscript. Changes made in the manuscript are marked using "track changes," and your questions are answered in this letter. We have also attached another clean version of our revised manuscript without track changes. We have addressed all minor comments from the reviewers. We have also addressed your main comment by adding the following lines to the manuscript (Lines 86-89).

*"This study builds on the previous work on socio-hydrology in transboundary rivers by Lu et al. (2021) and Wei et al. (2022) but distinguishes itself by incorporating the following additional important elements into our socio-hydrologic model: (1) the concept of social memory and quantitative components of C&C phenomena (e.g., political stability), (2) uncertainties in the representation of countries' decision making process, and (3) the heterogeneity of decision making across the riparian countries in their cooperation."*

Also, this study is consistent with the study by Wei et al. (2022) as it considers the concept of the social motives and power status in the model conceptualization (Figure 3) and equations (Eq1-3). The social motives of countries are shown in line 284-290 of the manuscript:

*"Consistent findings show that decision behaviors in resource dilemmas are mostly individualism (i.e., the drive to prioritize one's own interests) and competition (the drive to increase relative gains, the gap between one's achievement and that of the other) (Brewer & Kramer, 1986; Parks & Vu, 1994; Roch & Samuelson, 1997). Thus, here, countries' willingness to cooperate is evaluated based on their relative socio-economic gains (Kopelman et al., 2002; Mason, 2004)."*

As for the gained insights from the modeling the Eastern Nile River Basin, we summarized and discussed them in the section 5 by bullet points (Lines 521-550):

- *Ethiopia experienced two general trends in cooperation dynamics: a relatively high willingness to cooperate between 1983 and 2009 and a subsequent decrease. The model suggests that relative political stability and foreign direct investment can explain these two different phases, along with Ethiopia's food and energy gaps. The results of sensitivity analysis also show that energy and food gaps are the most important factors in Ethiopia's willingness to cooperate. This finding suggests that improvements in Ethiopia's food gap and energy gap can be a good motivation for Ethiopia for further negotiations in the basin. Also, the model suggest that a high level of Ethiopia's relative political stability and foreign direct investment can be a barrier to further basin cooperation. This is also supported by Mason (2004) who argued that the international economic and political asymmetry can negatively affect the basin cooperation.*

- *Sudan's willingness to cooperate dropped between 2003 and 2008, and recovered subsequently, the latter pattern likely reflecting the 2008 food crisis in Sudan. The importance of the agricultural sector was also shown in the result of sensitivity analysis, compared to other socio-political and hydrological factors in Sudan's willingness to cooperate. This result implies that Sudan is likely to be motivated in further negotiations in the basin by improvements in its food gap.*

- *The drop of Egypt's willingness to cooperate around 2007 appears to be related to negotiation deadlock while, later, Egypt entered a politically unstable phase and returned to the negotiation table, which is reflected by a recovery of its willingness to cooperate. Also, the result of sensitivity analysis showed the important role of political stability and the country's memory for Egypt's willingness to cooperate. Based on Egypt's past experience, it might be challenging for Egypt to fully trust upstream countries during further negotiations because Egypt is concerned about its historical water rights of the Nile as its main water resource. Thus, we suggest that building up Egypt's trust might be the very first step for any negotiations in the basin. Such trust might be strengthened by a basic commitment by all parties that a basin-wide agreement will be the basis of infrastructure, including GERD, operation.*

- *At the scale of the Eastern Nile Basin, the model highlights the role of trust and good memory from the past in increasing cooperation. For example, the increase in cooperation after 1984 brought the riparian countries together for the Promotion and Development of the Nile activity (TECCONILE) in 1992. Also, the result of sensitivity analysis showed that Ethiopia's food and energy gap and Sudan's food gap are the most important factors in the basin cooperation. These findings suggest that a further cooperative agreement should be more focused on improvements in Ethiopia's food and energy gap and Sudan's food gap while assuring no significance harm to Egypt's historical water use.*

**Reviewer 1:**

The quality of the manuscript has been largely improved. Only several minor comments:

1) In Figure 2, the authors introduces the C&C from 1959 to 2020. However, the study period was much shorter, the authors may briefly discuss the implications your results on the long historical period. It is noted that the input variables and results variables had inconsistent start year.

2) The authors may improve some sentences for example Line 94

3) Some references included in the text were not included in the reference list. Please check

Thank you for your time to review this paper. We have fixed all these issues in the manuscript. Specifically, the first comment has been addressed by the following (Lines 108-110):

*"This time period is longer than that of our modeling in section 3 as we need a long time period to investigate the socio-economic and political factors in the riparian countries' willingness to cooperate. It is worth mentioning that we did not use this long period for section 3 due to the lack of quantitative data."*

**Reviewer 2:**

Author's claims that this paper connects qualitative literature review with quantitative bases is not validated, especially considering their acknowledgements of a lack of sensitivity analysis in the final paragraph. Is this a methodological paper or a proof of a local phenomenon? It does not appear to be robust enough for both, yet that is how the writing is conducted. Do not state 'could be' without actually doing (in last paragraph, summarizing entire methodological approach).

Thank you for your time to review this paper. We do believe that the qualitative literature review and data paved the way for this study in two areas: (1) the qualitative literature review enabled us to investigate the socio-economic and political factors in the riparian countries' willingness to cooperate (section 2), and (2) the independent qualitative data enabled us to validate the overall trends of the model outputs. Also, this study provides a sensitivity analysis on the model parameters, which highlighted how surveying data can improve this modeling (section 3.5). The last paragraph in the conclusion states how this model can be improved by additional work. Specifically, one of them is sensitivity analysis on the model inputs and model structure (i.e., the equations of countries' decision making), which is different from our sensitivity analysis on model parameters. In fact, acknowledging the lack of these types of sensitivity analyses has nothing to do with the validation of this work.

We regret that these points were not clear in the text. Therefore, we modified the text in the last paragraphs and added new sentences to make our views and conclusions clear, as follows:

"*The ENSH model is built on the existing knowledge of the processes, but future studies could use alternative social theories and hypotheses, thereby improving our perceptual understanding of how the system works. In this regard, we should point out that our primary modeling purpose has been 'diagnostic learning' by simulating the past, complex behaviors pertaining to the conflict and cooperation in ENB – see the discussion in Razavi et al. (2022) for models as a tool for diagnostic learning. We believe this work, along with previous work cited in this paper, constitute the first steps towards building a predictive model for such phenomena to be used for future decision support.*"

"*Another potential source of uncertainty of the ENSH model is the input data (i.e., energy production, food consumption, food production, foreign direct investment, future reservoir storage, potential energy capacity, and relative political stability). The significance of such uncertainty in inputs on the outcome may be assessed through sensitivity analysis in future work.*"